# Stochastic Mirror Descent: Convergence Analysis and Adaptive Variants via the Mirror Stochastic Polyak Stepsize

**Ryan D'Orazio**                                                   *ryan.dorazio@mila.quebec*
*Mila, Université de Montréal*

**Nicolas Loizou**                                                          *nloizou@jhu.edu*
*Johns Hopkins University*

**Issam Laradji**                                                   *issam.laradji@gmail.com*
*ServiceNow Research*

**Ioannis Mitliagkas**                                           *ioannis@iro.umontreal.ca*
*Mila, Université de Montréal*
*Canada CIFAR AI Chair*

**Reviewed on OpenReview:** *https://openreview.net/forum?id=28bQiPWxHl*

## Abstract

We investigate the convergence of stochastic mirror descent (SMD) under interpolation in relatively smooth and smooth convex optimization. In relatively smooth convex optimization we provide new convergence guarantees for SMD with a constant stepsize. For smooth convex optimization we propose a new adaptive stepsize scheme — the mirror stochastic Polyak stepsize (mSPS). Notably, our convergence results in both settings do not make bounded gradient assumptions or bounded variance assumptions, and we show convergence to a neighborhood that vanishes under interpolation. Consequently, these results correspond to the first convergence guarantees under interpolation for the exponentiated gradient algorithm for fixed or adaptive stepsizes. mSPS generalizes the recently proposed stochastic Polyak stepsize (SPS) (Loizou et al., 2021) to mirror descent and remains both practical and efficient for modern machine learning applications while inheriting the benefits of mirror descent. We complement our results with experiments across various supervised learning tasks and different instances of SMD, demonstrating the effectiveness of mSPS.

## 1 Introduction

We consider the constrained stochastic optimization problem,

$$\min_{x \in \mathcal{X}} f(x) = \mathbb{E}_{\xi} \left[ f_{\xi}(x) \right], \tag{1}$$

where $\mathcal{X} \subseteq \mathbb{R}^d$ is a non-empty closed convex set that is possibly unbounded, and $\xi$ is a random vector supported on a set $\Xi$ such that $\mathbb{E}_{\xi} \left[ f_{\xi}(x) \right]$ is always well defined. We assume that it is possible to generate a sequence of independent and identically distributed (i.i.d.) realizations of $\xi$, and that for each $x \in \mathcal{X}$ $\mathbb{E}_{\xi} \left[ \nabla f_{\xi}(x) \right] = \nabla f(x)$. We use $\mathcal{X}_* \subset \mathcal{X}$ to denote the set of minimizers $x_*$ of (1) and assume that $\mathcal{X}_*$ is not empty.

A special case of interest is the finite-sum optimization problem where $\Xi = \{1, \cdots, n\}$ and $\mathbb{E}_{\xi} \left[ f_{\xi}(x) \right] = \sum_{i=1}^{n} \frac{f_i(x)}{n}$. Finite-sum optimization problems are often used in machine learning tasks where vector $x$

denotes the model parameters, $f_i(x)$ represents the loss on the training point $i$ and the goal is to minimize the average loss $f(x)$ across the training points while satisfying the problem constraints (expressed as $x \in \mathcal{X}$).

A common iterative approach to solve (1) when $\mathcal{X} = \mathbb{R}^d$ is stochastic gradient descent (SGD) (Robbins & Monro, 1951; Gower et al., 2019), iterates are updated in the negative direction of a gradient computed from a single realization of $\xi$. When the problem is constrained, $\mathcal{X} \subset \mathbb{R}^d$, one may employ projected methods such as stochastic projected gradient descent (SPGD). However, the convergence guarantees of both SGD and SPGD depend on values measured by the Euclidean norm. If the Euclidean structure is not naturally suited to the problem, then SGD and PGD can suffer a worse dependence on the dimension $d$. A powerful generalization of SGD and SPGD is stochastic mirror descent (SMD), permitting better convergence guarantees by matching the geometry of the problem (Nemirovski & Yudin, 1983; Beck & Teboulle, 2003). For example in some cases, SMD can improve SGD's $\sqrt{d}$ dependence to $\sqrt{\log(d)}$ (Ben-Tal et al., 2001; Beck & Teboulle, 2003). Furthermore, mirror descent leverages non-Euclidean projections, allowing for different choices of projections that are perhaps better suited to the constraint set. For example, in sequential games particular instances of mirror descent have been designed to allow for efficient projections on the strategy spaces of players (Hoda et al., 2010; Kroer et al., 2020).

A classical analysis of mirror descent and first-order methods often relies on smoothness with respect to some norm $||\cdot||$. The norm $||\cdot||$ is often used in selecting the appropriate instance of mirror descent (Dekel et al., 2012; Bubeck, 2015). However, a recent trend is to study non-euclidean methods like mirror descent with the more general assumption of relative smoothness (Birnbaum et al., 2011; Bauschke et al., 2017; Lu et al., 2018). Several applications of interest are not smooth but relatively smooth, *e.g.*, algorithmic game theory (Birnbaum et al., 2011); Poisson inverse problems (Bertero et al., 2009); and more (Lu et al., 2018).

In contrast to deterministic methods, stochastic methods under relative smoothness have received less attention. We contribute to the literature of SMD with new constant-stepsize results under relative smoothness and new adaptive-stepsize results under smoothness, both of which use weak assumptions on the noise.

## 1.1 Main Contributions

The key contributions of this work are as follows:

- **Technical assumptions on the noise.** Unlike most of the SMD literature, all our convergence results, with relative smoothness or smoothness, and for fixed and adaptive stepsizes, do not make bounded gradient or bounded variance assumptions. Instead we use the *finite optimal objective difference*, introduced by Loizou et al. (2021), for our adaptive smooth setting and introduce a new constrained version for the relative smooth setting. More precisely, Loizou et al. (2021) assume

$$\sigma^2 := f(x_*) - \mathbb{E}_\xi \left[ f_\xi^* \right] < \infty, \tag{2}$$

where $f(x_*) = \min_{x \in \mathcal{X}} f(x)$, $f_\xi^* := \inf_{x \in \mathbb{R}^d} f_\xi(x)$. In the finite-sum case, $\mathbb{E}\left[ f_i^* \right] = \sum_{i=1}^n \frac{f_i^*}{n}$. For our relative smooth results we introduce the *constrained finite optimal objective difference*, a refinement that depends on the constraint $\mathcal{X}$,

$$\sigma_{\mathcal{X}}^2 := f(x_*) - \mathbb{E}_\xi \left[ f_\xi^*(\mathcal{X}) \right] < \infty, \tag{3}$$

where $f_\xi^*(\mathcal{X}) = \inf_{x \in \mathcal{X}} f_i(x)$ and in the finite-sum case $\mathbb{E}\left[ f_i^*(\mathcal{X}) \right] = \sum_{i=1}^n \frac{\inf_{x \in \mathcal{X}} f_i(x)}{n}$. From definition we have that the constrained version is a weaker assumption than its unconstrained counterpart since $\sigma_{\mathcal{X}}^2 \leq \sigma^2$.

- **Novel adaptive SMD.** We propose the mirror stochastic Polyak stepsize (mSPS) as an adaptive stepsize for SMD. Contrary to most adaptive mirror descent methods for stochastic optimization we do not use an online to batch reduction (Cesa-Bianchi et al., 2004; Littlestone, 1989). Hence, we avoid common assumptions like bounded constraints and provide efficient convergence results like linear convergence under strong convexity and smoothness.

- **Exact convergence with interpolation.** In modern machine learning, overparametrized models capable of driving training error to zero have increasingly become important in both theory and

in practice (Ma et al., 2018; Zhang et al., 2021). Under these conditions (see Definition 4) it has been shown that SGD enjoys favourable guarantees with exact convergence (Ma et al., 2018). Our analysis with $\sigma^2$, and $\sigma^2_{\mathcal{X}}$, shows that SMD inherits similar guarantees with fast and exact convergence under interpolation. We are unaware of other similar results for SMD with adaptive stepsizes. Moreover, our results provide the *first* convergence guarantees under interpolation for the exponentiated gradient algorithm under both the relative smoothness and the classic smoothness settings.

- **Extensive numerical experiments for adaptive SMD.** We demonstrate the adaptive capability of our proposed adaptive stepsize across a wide variety of domains and mirror descent algorithms for both constrained and unconstrained problems.

## 2 Related Work

**Stochastic Mirror Descent.** SMD is often analyzed as a stochastic method for optimizing non-smooth Lipschitz continuous convex functions (Nemirovski et al., 2009; Bubeck, 2015; Beck, 2017). These results can be derived from online to batch reductions yielding a $O(1/\sqrt{T})$ convergence rate (Cesa-Bianchi et al., 2004; Duchi et al., 2010; Orabona, 2019). In the case of non-smooth and strongly convex, several works improve the results following from online regret bounds with a $O(1/T)$ convergence rate (Hazan & Kale, 2014; Ghadimi & Lan, 2012; Juditsky & Nesterov, 2010). Under smoothness, similar improvements can be made (Dekel et al., 2012; Bubeck, 2015). All of these results use bounded variance or bounded gradient assumptions.[1] These assumptions can be difficult to verify, and may impose further restrictions. For example, one cannot in general assume a bounded gradient with strong convexity if $\mathcal{X}$ is unbounded, therefore it is common to assume $\mathcal{X}$ is compact.

In relative smooth optimization Hanzely & Richtarik (2021) make an assumption similar to bounded variance. More recently, Dragomir et al. (2021) avoid making bounded variance or bounded gradient assumptions under relative smoothness but make larger restrictions on the class of problems and mirror descent methods. We make an in-depth comparison with these works in Section 5. We also add that there are several works related to randomized coordinate descent methods (Hanzely & Richtarik, 2021; Gao et al., 2020; Hendrikx et al., 2020), and with variance reduction (Hendrikx et al., 2020; Dragomir et al., 2021).

**Interpolation in constrained optimization.** Interpolation conditions have mostly been studied with SGD in unconstrained settings (Gower et al., 2019; Vaswani et al., 2019a) or with SMD and conditions that do not incorporate constraints Hanzely & Richtarik (2021). Consequently, these conditions can yield large or unbounded neighborhoods of convergence in constrained optimization. Xiao et al. (2022) addresses some of these shortcomings by introducing the variance based weak growth condition to model interpolation under stochastic constrained optimization. However, the condition only holds under interpolation and requires the variance to be zero at the optimum. In comparison, our constraint-aware condition $\sigma^2_{\mathcal{X}}$ can hold without interpolation and does not require variance to be zero at the optimum.

**Adaptive stepsizes.** Adaptive stepsizes for mirror descent have a long history. Accumulating past gradients or subgradients to set a stepsize, $\eta_t \propto 1/\sqrt{\sum_{s=1}^t \|g_s\|_*^2}$ , can be traced back to online learning (Auer et al., 2002; Streeter & McMahan, 2010). Recently, similar coordinate-wise stepsizes such as ADAGRAD (McMahan & Streeter, 2010; Duchi et al., 2011) have been proposed. The convergence guarantees for these methods in convex optimization use online regret bounds, requiring sublinear regret. Unfortunately, all mirror descent methods with the aforementioned stepsizes require a bounded constraint; when the problem is unconstrained Orabona & Pál (2018) prove a $\Omega(T)$ worst case lower bound for the regret.[2] Furthermore, in the stochastic case, bounded gradient and variance assumptions are made when using the online to batch reduction (Duchi, 2018; Orabona, 2019). In contrast our methods employ a completely different stepsize and we make a very

---

[1] Lei & Tang (2018) derive results for non-smooth and strongly convex functions without bounded subgradients but assume a weak growth condition.

[2] Convergence results may still be possible without online to batch reductions; for example, in the case of unconstrained SGD see Li & Orabona (2019).

weak assumption on the noise. Another line of related work includes adaptive stepsizes for mirror descent with non-smooth functional constraints (Bayandina, 2017; Bayandina et al., 2018; Stonyakin et al., 2019).

**Polyak stepsize.** Our adaptive stepsizes are in the spirit of Polyak's stepsize — originally proposed for deterministic projected subgradient descent (Polyak, 1987). In the deterministic setting Polyak's results have been been successfully extended and used for solving weakly convex and smooth problems (Boyd et al., 2003; Davis et al., 2018; Hazan & Kakade, 2019). More recently, variations of the Polyak stepsize have been proposed for stochastic optimization (Loizou et al., 2021; Prazeres & Oberman, 2021; Berrada et al., 2020; Gower et al., 2021). The adaptive stepsizes proposed herein are a generalization of SPS proposed and analyzed by Loizou et al. (2021) (see Section 4.2) to the constrained case and for mirror descent.

## 3 Background

We denote vectors within the feasible set as $x \in \mathcal{X} \subseteq \mathbb{R}^d$, where $\mathbb{R}$ is the set of real numbers. We use the subscript to denote time, after $t$ time steps the average of the iterates $x_1, \cdots, x_t$ is $\bar{x}_t = 1/t \sum_{s=1}^t x_s$. With a slight abuse of notation we may also refer to the $i$th coordinate of $x$ as $x_i$, $x = (x_1, \cdots, x_d)$. Whether the subscript refers to time or the coordinate is clear from context. We denote $||\cdot||_2$ as the Euclidean norm and $||\cdot||$ as any arbitrary norm with corresponding dual norm $||x||_* = \sup_y\{\langle x, y \rangle : ||y|| \leq 1\}$.

For a differentiable function $\psi$, we define the difference between $\psi(x)$ and the first order approximation of it at $y$ as the Bregman divergence $B_\psi(x; y)$.

**Definition 1** (Bregman divergence). *Let $\psi : \mathcal{D} \to \mathbb{R}$ be differentiable on* $\text{int}\,\mathcal{D}$. *Then the Bregman divergence with respect to $\psi$ is $B_\psi : \mathcal{D} \times \text{int}\,\mathcal{D} \to \mathbb{R}$, defined as*

$$B_\psi(x; y) = \psi(x) - \psi(y) - \langle \nabla \psi(y), x - y \rangle.$$

A differentiable function $f$ is convex on a convex set $\mathcal{X}$ if $B_f(x; y) \geq 0$ for any $x, y \in \mathcal{X}$. Similarly, a function $f$ is $L$-smooth with respect to a norm $||\cdot||$ if $B_f(x; y) \leq \frac{L}{2} ||x - y||^2$, and is $\mu$-strongly convex with respect to the norm $||\cdot||$ if $\frac{\mu}{2} ||x - y||^2 \leq B_f(x; y)$.

We will also refer to the generalization of smoothness and strong convexity — relative smoothness and relative strong convexity defined below.

**Definition 2.** *A function $f$ is $L$-smooth relative to $\psi$ on $\mathcal{X}$ if for all $(x, y) \in \mathcal{X} \times (\mathcal{X} \cap \text{int}\,\mathcal{D})$ it holds that: $B_f(x; y) \leq L B_\psi(x; y)$.*

**Definition 3.** *A function $f$ is $\mu$-strongly convex relative to $\psi$ on $\mathcal{X}$ if for all $(x, y) \in \mathcal{X} \times (\mathcal{X} \cap \text{int}\,\mathcal{D})$ it holds that: $\mu B_\psi(x; y) \leq B_f(x; y)$.*

### 3.1 Mirror descent

To solve problem (1) we consider the general stochastic mirror descent update with a convex function $\psi$ and domain $\mathcal{D}$

$$x_{t+1} = \arg\min_{x \in \mathcal{X}} \langle \nabla f_{\xi_t}(x_t), x \rangle + \frac{1}{\eta_t} B_\psi(x; x_t). \tag{4}$$

Where $\xi_t$ is a realization of $\xi$ that is i.i.d.. In the non-smooth or deterministic setting $\nabla f_{\xi_t}(x_t)$ may be replaced by a subgradient or the full gradient respectively. To make the updates well defined, all we require is that $x_{t+1} \in \text{int}\,\mathcal{D}$ in update (4), otherwise $B_\psi(\cdot, x_{t+1})$ will be undefined at the next step.

**Assumption 1.** *Let $\psi$ be convex with domain $\mathcal{D}$, differentiable over* $\text{int}\,\mathcal{D}$, *and $\mathcal{X} \subseteq \mathcal{D}$. For any $g$, and any stepsize $\eta_t > 0$, $x_{t+1} = \arg\min_{x \in \mathcal{X}} \langle g, x \rangle + \frac{1}{\eta_t} B_\psi(x; x_t) \in \text{int}\,\mathcal{D}$.*

For example the following assumption by Orabona (2019) would be sufficient.

**Assumption** (Section 6.4 Orabona (2019))**.** *Let $\psi : \mathcal{D} \to \mathbb{R}$ be a strictly convex function such that $\mathcal{X} \subseteq \mathcal{D}$, we require either one of the following to hold:* $\lim_{x \to \partial \mathcal{X}} ||\nabla \psi(x)||_2 = +\infty$ *or* $\mathcal{X} \subseteq \text{int } \mathcal{D}$.[3]

The first requirement from Orabona (2019) amounts to assuming $\psi$ is a Legendre function (*i.e.*, essentially smooth and strictly convex), which implies that $x_{t+1} \in \text{int } \mathcal{D}$ (Cesa-Bianchi & Lugosi, 2006). Otherwise, if the second condition holds then the update is also well defined. Furthermore, we note that other assumptions can be made to guarantee $x_{t+1} \in \text{int } \mathcal{D}$; for more examples see Bauschke et al. (2003).

Another common assumption is to assume $\psi$ is strongly convex over $\mathcal{X}$, which will be important for our adaptive stepsize in Section 6, however, it is not needed for the constant step size results of Section 5.

The following is a standard one step mirror descent lemma and will be used often (Beck, 2017; Bubeck, 2015; Orabona, 2019; Duchi, 2018), this particular statement and proof is taken from Lemma 6.7 by Orabona (2019) and we include the full proof in the appendix for completeness. All other omitted proofs are deferred to the appendix.

**Lemma 1.** *Let $B_\psi$ be the Bregman divergence with respect to a convex function $\psi : \mathcal{D} \to \mathbb{R}$ and assume assumption 1 holds. Let $x_{t+1} = \arg \min_{x \in \mathcal{X}} \langle g_t, x \rangle + \frac{1}{\eta_t} B_\psi(x; x_t)$. Then for any $x_* \in \mathcal{X}$*

$$B_\psi(x_*; x_{t+1}) \leq B_\psi(x_*; x_t) - \eta_t \langle g_t, x_t - x_* \rangle - B_\psi(x_{t+1}; x_t) + \eta_t \langle g_t, x_t - x_{t+1} \rangle. \tag{5}$$

*Furthermore if $\psi$ is $\mu_\psi$-strongly convex over $\mathcal{X}$ then*

$$B_\psi(x_*; x_{t+1}) \leq B_\psi(x_*; x_t) - \eta_t \langle g_t, x_t - x_* \rangle + \frac{\eta_t^2}{2\mu_\psi} ||g_t||_*^2. \tag{6}$$

The SMD update (4) recovers both SGD and SPGD if $\psi$ is taken to be $\frac{1}{2} ||\cdot||_2^2$. Some other interesting examples include the case where $\psi(x) = \frac{1}{2} ||x||_p^2$ for $1 < p \leq 2$ (Grove et al., 2001; Gentile, 2003). If we instead use $\psi(x) = \frac{1}{2} \langle x, Mx \rangle = \frac{1}{2} ||x||_M^2$ for a positive definite matrix $M$ then we recover the scaled projected gradient algorithm, $x_{t+1} = \arg \min_{x \in \mathcal{X}} ||x_t - \eta_t M^{-1} g_t - x||_M^2$ (Bertsekas & Tsitsiklis, 2003). Another common setup is when $\psi$ is taken to be the negative entropy with a constraint set $\mathcal{X} = \Delta^d = \{x_i \geq 0 | \sum_{i=1}^d x_i = 1\}$. In this case $\psi$ is 1-strongly convex with respect to $||\cdot||_1$ and the update rule corresponds to the exponentiated gradient algorithm (Littlestone & Warmuth, 1994; Kivinen & Warmuth, 1997; Beck & Teboulle, 2003; Cesa-Bianchi & Lugosi, 2006).

### 3.2 Overparameterization, interpolation, and constrained interpolation

Modern machine learning models are expressive and often over-parametrized, i.e., they can fit or interpolate the training dataset (Zhang et al., 2021). For example when problem (1) is the training problem of an over-parametrized model such as a deep neural network (Ma et al., 2018) or involves solving a consistent linear system (Loizou & Richtárik, 2020b;a) or problems such as deep matrix factorization (Rolinek & Martius, 2018; Vaswani et al., 2019b), each individual loss function $f_i$ attains its minimum at $x_*$. That is the following interpolation condition is satisfied.

**Definition 4** (Interpolation)**.** *We say that the interpolation condition holds when there exists $x_* \in \mathcal{X}_*$ such that $f_\xi(x_*) = \inf_{x \in \mathbb{R}^d} f_\xi(x)$ almost surely.*

In the finite-sum setting this condition amounts to $f_\xi(x_*) = \inf_{x \in \mathbb{R}^d} f_\xi(x)$ for all $i \in \{1, \cdots, n\}$. Note that when the interpolation condition is satisfied, it follows that $\sigma^2 = 0$ (see (2)).

The use of interpolation in the literature has mostly been discussed within the context of unconstrained optimization. Despite Definition 4 being designed to study SGD for unconstrained optimization, we show that constrained optimization with mirror descent enjoys similar convergence benefits if $\sigma^2 = 0$. However, the standard definition of interpolation, as given by Definition 4, does not adequately describe interpolation with respect to the constraint $\mathcal{X}$. For example, consider the case where $f_i(x) = f(x)$ for all $x \in \mathcal{X}$ but do not

---

[3]$\partial \mathcal{X}$ denotes the boundary of $\mathcal{X}$.

agree outside the constraint $\mathcal{X}$. In this case, it is possible to have $\sigma^2$ arbitrarily large, meanwhile, there is no variance in the stochastic gradient $\mathbb{E}\left[||\nabla f(x) - \nabla f_i(x)||_2^2\right] = 0$ – rendering the problem non-stochastic.

Xiao et al. (2022) describe an interpolation-like condition within constraint optimization, however, this condition requires the stochastic gradient to have zero variance at the optimum, which need not hold generally despite all $f_\xi$ sharing a common minimum (see for example Figure 1). Therefore, we also make use of the following constrained interpolation condition:

**Definition 5** (Contrained Interpolation). *We say that the interpolation condition holds with respect to the constraint $\mathcal{X}$ if there exists $x_* \in \mathcal{X}_*$ such that $f_\xi(x_*) = \inf_{x \in \mathcal{X}} f_\xi(x)$ almost surely.*

Similar to Definition 4 we have interpolation with respect to $\mathcal{X}$ holds when $\sigma_{\mathcal{X}}^2 = 0$. In the finite-sum setting, constrained interpolation reduces to $f_i(x_*) = \inf_{x \in \mathcal{X}} f_i(x)$ for all $i \in \{1, \cdots, n\}$.

## 4 Constant and Polyak stepsize for mirror descent

In this section we provide background on constant stepsize selection for mirror descent. For non-constant setpsize, we introduce our natural extensions of the classic Polyak stepsize and SPS for mirror descent.

### 4.1 Constant Stepsize

When a function is $L$-smooth with respect to the Euclidean norm, a common stepsize for gradient descent is $\eta = 1/L$, allowing for convergence in many settings (Bubeck, 2015). Similarly for an $L$-relatively smooth function with respect to $\psi$, the prescibed stepsize for mirror descent using $\psi$ is $\eta = 1/L$ (Birnbaum et al., 2011; Lu et al., 2018). In the stochastic and relatively smooth case, Hanzely & Richtarik (2021) use $\eta = 1/L$ as well as different stepsize schedules. In Section 5, we provide new convergence guarantees (under weaker assumptions) for SMD with $\eta = 1/L$.

### 4.2 Polyak Stepsize

An alternative method to selecting a stepsize, as suggested by Polyak (Polyak, 1987), is to take $\eta_t$ by minimizing an upper bound on $||x_{t+1} - x_*||_2^2$. From Lemma 1, if we take $\psi = \frac{1}{2}||\cdot||_2^2$ and assume $g_t \in \partial f(x_t)$ is a subgradient at $x_t$ for $f$ then we recover a well known inequality for projected subgradient descent[4]

$$\frac{1}{2}||x_* - x_{t+1}||_2^2 \le \frac{1}{2}||x_* - x_t||_2^2 - \eta_t(f(x_t) - f(x_*)) + \frac{\eta_t^2}{2}||g_t||_2^2.$$

Minimizing the right hand size with respect to $\eta_t$ yields Polyak's stepsize, $\eta_t = {}^{(f(x_t) - f(x_*))}/{||g_t||_2^2}$ (Polyak, 1987; Beck, 2017). Following in a similar fashion, we propose a generalization of Polyak's stepsize for mirror descent. If $\psi$ is $\mu_\psi$-strongly convex[5] with respect to the norm $||\cdot||$ then we can minimize the right hand side of equation (6) to arrive at the mirror Polyak stepsize

$$\eta_t = \frac{\mu_\psi(f(x_t) - f(x_*))}{||g_t||_*^2}. \tag{7}$$

Despite the well-known connection between projected subgradient descent and mirror descent (Beck & Teboulle, 2003), this generalization of Polyak's stepsize is absent from the literature. For completeness, we include analysis of the non-smooth case in Section D of the appendix, including both a $O(1/\sqrt{t})$ convergence and a last iterate convergence result. As expected, mirror descent with the mirror Polyak stepsize maintains the benefits of mirror descent — it permits a mild dependence on the dimension of the space.

---

[4] After using the fact that $g_t$ is a subgradient, $f(x_t) - f(x_*) \le \langle g_t, x_t - x_* \rangle$.
[5] Without loss of generality we could assume $\psi$ to be 1-strongly convex and scale $\psi$ by $1/\mu_\psi$. The stepsize would remain the same, scaling of $\psi$ inversely scales the stepsize.

However, it inherits the impractical issues with the Polyak stepsize — knowledge of $f(x_*)$ and an exact gradient or subgradient.

In the stochastic setting Loizou et al. (2021) propose the more practical stochastic Polyak stepsize (SPS), $\eta_t = (f_{\xi_t}(x_t) - f_{\xi_t}^*)/c||\nabla f_{\xi_t}(x_t)||_2^2$, and the bounded variant $\text{SPS}_{\max}$, $\eta_t = \min\{(f_{\xi_t}(x_t) - f_{\xi_t}^*)/c||\nabla f_{\xi_t}(x_t)||_2^2, \eta_b\}$. Where $f_{\xi_t}^*$ is known in many machine learning applications, and $c$ is a scaling parameter that depends on the class of functions being optimized (Loizou et al., 2021).

Similar to our generalization of Polyak's stepsize (7), we propose a generalization of SPS and $\text{SPS}_{\max}$ for mirror descent, the mirror stochastic Polyak stepsize (mSPS) and the bounded variant $\text{mSPS}_{\max}$,

$$\text{mSPS}: \eta_t = \frac{\mu_\psi(f_{\xi_t}(x_t) - f_{\xi_t}^*)}{c||\nabla f_{\xi_t}(x_t)||_*^2}, \tag{8}$$

$$\text{mSPS}_{\max}: \eta_t = \min\left\{\frac{\mu_\psi(f_{\xi_t}(x_t) - f_{\xi_t}^*)}{c||\nabla f_{\xi_t}(x_t)||_*^2}, \eta_b\right\}. \tag{9}$$

### 4.2.1 Self-bounding property of mSPS

An important property of SPS and mSPS is its self-bounding property for when $f_{\xi_t}$ is $L$-smooth and $\mu$-strongly convex with respect to a norm $||\cdot||$,

$$\frac{\mu_\psi}{2cL} \leq \eta_t = \frac{\mu_\psi(f_{\xi_t}(x_t) - f_{\xi_t}^*)}{c||\nabla f_{\xi_t}(x_t)||_*^2} \leq \frac{\mu_\psi}{2c\mu}. \tag{10}$$

We extensively use the lower bound, also known as the self-bounding property of smooth functions (Srebro et al., 2010), and we provide a complete proof in the appendix (Section E). A proof of the upper bound can be found in Orabona (2019)[Corollary 7.6].

## 5 Convergence with constant stepsize in relatively smooth optimization

In this section we provide new convergence results for SMD with constant stepsize under relative smoothness. We provide the following lemma which allows us to bound the last two terms in (5). This result can be seen as a generalization of Lemma 2 in Collins et al. (2008), where the exponentiated gradient algorithm is studied under the relative smoothness assumption.

**Lemma 2.** *Suppose $f$ is $L$-smooth relative to $\psi$. Then if $\eta \leq \frac{1}{L}$ we have*

$$-B_\psi(x_{t+1}; x_t) + \eta\langle\nabla f(x_t), x_t - x_{t+1}\rangle \leq \eta(f(x_t) - f(x_{t+1})).$$

### 5.1 Relative smoothness and strong convexity

For an appropriately selected stepsize, we have that SMD enjoys a linear rate of convergence to a neighborhood of the minimum $x_*$.

**Theorem 1.** *Assume $\psi$ satisfies assumption 1. Furthermore assume $f$ to be $\mu$-strongly convex relative to $\psi$ over $\mathcal{X}$, and $f_\xi$ to be $L$-smooth relative to $\psi$ over $\mathcal{X}$ almost surely. Then SMD with stepsize $\eta \leq \frac{1}{L}$ guarantees*

$$\mathbb{E}\left[B_\psi(x_*; x_{t+1})\right] \leq (1 - \mu\eta)^t B_\psi(x_*; x_1) + \frac{\sigma_{\mathcal{X}}^2}{\mu}.$$

Importantly, we do not assume $f_\xi$ to be convex and under interpolation with respect $\mathcal{X}$ we have $\sigma_{\mathcal{X}}^2 = 0$ implying SMD will converge to the true solution if $\psi$ is strictly convex. If $\psi$ is strongly convex then Theorem 1 provides a linear rate on the expected distance $||x_{t+1} - x_*||^2$ for some norm $||\cdot||$. For example Collins et al. (2008) show that a particular loss $f_\xi$ appearing in the dual problem of fitting regularized log-linear

models is both smooth and strongly convex relative to the negative entropy function. In this case, our results provide a linear rate on $\mathbb{E}\left[||x_{t+1} - x_*||_1^2\right]$ for the stochastic exponentiated gradient algorithm since $\psi$ (negative entropy) is strongly convex with respect to the norm $||\cdot||_1$.

In the case of interpolation, we have that $B_\psi(x_*; x_{t+1}) \to 0$ almost surely. If $\psi$ is strictly convex then $x_t \to x_*$ almost surely.

**Corollary 2.** *Under the same assumptions as Theorem 1, if $\sigma_{\mathcal{X}}^2 = 0$ then $B_\psi(x_*; x_{t+1}) \to 0$ almost surely.*

### 5.2 Relative smoothness without convexity

Similar to Theorem 1 we show convergence of a quantity to a neighborhood, only assuming $f_\xi$ to be $L$-smooth relative to $\psi$, where $f$ or $f_\xi$ need not be convex.

**Theorem 3.** *Assume $\psi$ satisfies assumption 1. Furthermore assume $f_\xi$ to be $L$-smooth relative to $\psi$ over $\mathcal{X}$ almost surely. Then SMD with stepsize $\eta \leq \frac{1}{L}$ guarantees*

$$\mathbb{E}\left[\frac{1}{t}\sum_{s=1}^{t} B_f(x_*; x_s)\right] \leq \frac{B_\psi(x_*; x_1)}{\eta t} + \sigma_{\mathcal{X}}^2.$$

The above guarantee also implies a result for the "best" iterate if $f$ is convex, $\mathbb{E}\left[\min_{1 \leq s \leq t} B_f(x_*; x_s)\right]$, to a neighborhood. If $f$ is strictly convex then this implies at least one iterate $x_s$ converges to a neighborhood of $x_*$ in expectation. If $f$ is strictly convex and 1-coercive[6] then its conjugate function $f^*$ is also strictly convex (Hiriart-Urruty & Lemaréchal, 2004)[Corollary 4.1.3] and we have $B_f(x_*; x_s) = B_{f^*}(\nabla f(x_s); \nabla f(x_*))$ (Bauschke et al., 1997)[Theorem 3.7], implying that the average gradients $1/t \sum_{s=1}^{t} \nabla f(x_s)$ or at least one of $\nabla f(x_s)$ converges to a neighborhood of $\nabla f(x_*)$. If $f$ happens to be convex and $L$-smooth with respect to a norm $||\cdot||$ then $\frac{1}{2L}||\nabla f(x_*) - \nabla f(x_s)||_*^2 \leq B_f(x_*; x_s)$(Nesterov, 2018)[Theorem 2.1.5], providing a similar convergence guarantee on the distance of gradients to $\nabla f(x_*)$.

Similar to Theorem 1, an almost surely convergence result follows from Theorem 3 under interpolation.

**Corollary 4.** *Under the assumptions of Theorem 3, if $f$ is convex and $\sigma_{\mathcal{X}}^2 = 0$ then $B_f(x_*; x_t) \to 0$ almost surely.*

#### 5.2.1 Application of Theorem 3, solving linear systems

Theorem 3 provides convergence for the unconventional quantity $B_f(x_*; x_t)$, however, this quantity is sometimes equal to $f(x_t) - f(x_*)$, as in the case of solving linear systems. In this case Theorem 3 automatically gives a result for the quantity $\mathbb{E}[f(\bar{x}_t) - f(x_*)]$ if $f$ is convex.

More formally, solving a constrained linear system amounts to to finding $x_*$ such that

$$Ax_* = b, \text{ and } x_* \in \mathcal{X}. \tag{11}$$

Problem (11) can be reformulated as a constrained finite sum problem with $f_i(x) = \frac{1}{2}(\langle A_{i:}, x \rangle - b_i)^2$, where $A_{i:}$ and $b_i$ denote the $i^{th}$ row and component of $A$ and $b$ respectively. Note that $x_*$ interpolates all $f_i$ since $f_i(x_*) = 0$ by construction, with $\nabla f_i(x_*) = 0$ and $\sigma_{\mathcal{X}}^2 = 0$. Since the divergence $B_{f_i}$ is symmetric[7] and $B_f$ is simply the average over $B_{f_i}$ it holds that

$$B_f(x_*; x_t) = \sum_{i=1}^{n} \frac{B_{f_i}(x_*; x_t)}{n} = \sum_{i=1}^{n} \frac{B_{f_i}(x_t; x_*)}{n} = \sum_{i=1}^{n} \frac{f_i(x_t) - f_i(x_*)}{n} = f(x_t) - f(x_*).$$

Where the third equality follows from the fact that $\nabla f_i(x_*) = 0$.

Theorem 3 therefore gives a convergence result for the gap $\mathbb{E}[f(\bar{x}_t) - f(x_*)]$, and Corollary 4 guarantees $f(x_t) \to f(x_*)$, provided each $f_i$ is relatively smooth. Relative smoothness of $f_i$ holds if $\psi$ is strongly convex

---

[6] A function is 1-coercive if $\lim_{||x|| \to \infty} f(x)/||x|| = +\infty$.

[7] See Proposition 1 in the appendix for a proof.

since for any norm $||\cdot||$ there exists a constant $L_i$ for which $f_i$ is $L_i$-smooth with respect to $||\cdot||$. Therefore, taking $L = \max_i L_i$ gives

$$B_{f_i}(x;y) \leq \frac{L}{2}\,||x-y||^2 = \frac{L\mu_\psi}{2\mu_\psi}\,||x-y||^2 \leq \frac{L}{\mu_\psi}B_\psi(x;y).$$

### 5.2.2 EG for finding stationary distributions of Markov chains

An important example of problem (11) for which $x_*$ exists and $\mathcal{X} \neq \mathbb{R}^d$ is the problem of finding a stationary distribution of a Markov chain with transition matrix $P$. That is, to find $x_*$ such that

$$(P^\top - I)x_* = 0, \text{ and } x_* \in \Delta^n. \tag{12}$$

Problem (12) is ubiquitous in science and machine learning, for example in online learning many algorithms require computing a stationary distribution of a Markov chain at each iteration(Greenwald et al., 2006; Blum & Mansour, 2007).

From the above discussion we can formulate the problem (12) as a finite-sum problem with constraint $\mathcal{X} = \Delta^m$. A natural choice for the simplex constraint is the stochastic EG algorithm (SMD with $\psi$ taken to be negative entropy). Denoting the $g_i = (P^\top - I)_{i:}$ we have that $f_i(x) = \frac{1}{2}\langle g_i, x\rangle^2$, and is 1-smooth with respect to $||\cdot||_1$. Since negative entropy is 1-strongly convex with respect to $||\cdot||_1$ we have that $f_i$ is also 1-smooth relative to $\psi$. Therefore, Theorem 3 and Corollary 4 guarantee $\mathbb{E}\left[f(\bar{x}) - f(x_*)\right]$ and $f(x_t) - f(x_*)$ converge to zero, respectively, for the following stochastic update: sample $i \in \{1, \cdots, n\}$ uniformly at random and

$$y_{t+1} = x_t \odot \exp\left(-\nabla f_i(x_t)\right), \; x_{t+1} = \frac{y_{t+1}}{||y_{t+1}||_1}. \tag{13}$$

Where $\odot$ and exp are component wise multiplication and component wise exponentiation respectively.

We highlight that no other existing works show convergence without a neighborhood under interpolation for the EG algorithm. Additionally, problem (12) also exhibits a natural occurence where $f_i^*$ is known and equal to 0, rendering mSPS (8) computable. In Section 6, we demonstrate similar guarantees with mSPS.

### 5.3 Comparison with related works

In the constant stepsize and relatively smooth regime, Hanzely & Richtarik (2021) and Dragomir et al. (2021) provide convergence guarantees for SMD under different assumptions and to different neighborhoods. We provide an in-depth comparison as well as demonstrate via an example where convergence to the solution is not guaranteed by previous works but is possible by Theorem 1.

Hanzely & Richtarik (2021) make an assumption akin to bounded variance, they assume $\mathbb{E}\left[\langle\nabla f(x_t)-\nabla f_{\xi_t}(x_t), x_{t+1}-\tilde{x}_{t+1}\rangle|x_t\right]/\eta \leq \sigma^2$, where $\tilde{x}_{t+1}$ is the mirror descent iterate using the true gradient $\nabla f(x_t)$. In the case of relative smoothness and relative strong convexity they show a linear rate of convergence to a neighborhood for $\mathbb{E}\left[f(\bar{x}_t) - f(x_*)\right]$ (Theorem 5.3), where $\bar{x}_t$ is a particular weighted average of the iterates $(x_1, \cdots, x_t)$. Without relative strong convexity, a similar result is shown for the uniform average $\bar{x}_t$ with a rate of $O(1/T)$ using a particular schedule of stepsizes (Corollary 5.5).

Our results more closely resemble the work of Dragomir et al. (2021), giving the same rates and exact convergence with interpolation (Theorem 1 and Theorem 3), however, there are several *important* differences. Firstly, our results apply to a wider range of problems and mirror descent methods. Dragomir et al. (2021) do not decouple the domain $\mathcal{D}$ of the function $\psi$ and the constraint set $\mathcal{X}$, they assume that $x_{t+1} \in \text{int } \mathcal{X}$ where $\nabla\psi(x_{t+1}) = \nabla\psi(x_t) - \eta_t\nabla f_{\xi_t}(x_t)$. Therefore their definition of mirror descent *precludes* the famous exponentiated gradient algorithm or projected gradient descent—no projection steps are allowed in their definition. Furthermore, our analysis allows $x_*$ to be anywhere in $\mathcal{X}$ while Dragomir et al. (2021) require $\nabla f(x_*) = 0$ and exclude the case when $x_*$ is on the boundary of $\mathcal{X}$. Secondly, our neighborhoods of convergence are different, they show convergence to a different neighborhood $\eta\sigma^2/\mu$ ($\eta\sigma^2$ in the smooth case),

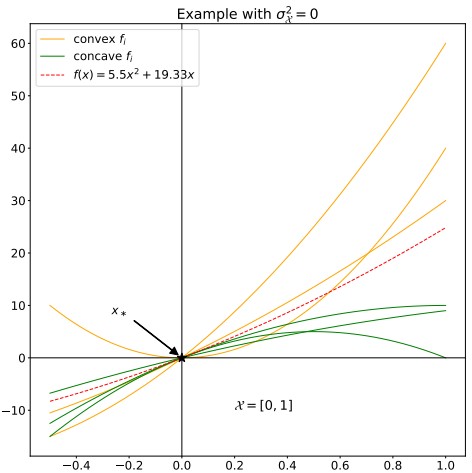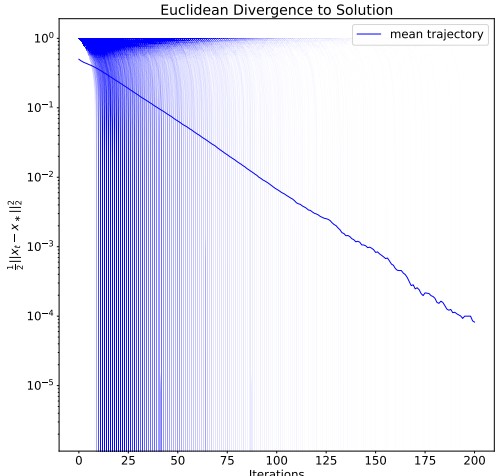

Figure 1: Finite-sum example of Theorem 1 with SPGD and $\psi = {}^1\!/\!{}_2\,\|\cdot\|_2^2$. (left) $f(x)$ is strongly convex and is a sum of smooth $f_i$ that are either non-convex or strongly convex functions. (right) As predicted by Theorem 1, linear convergence is observed for the mean trajectory of SPGD over 10,000 runs.

where $\sigma^2$ is such that $\mathbb{E}\left[\|\nabla f_\xi(x_*)\|_{\nabla^2\psi^*(z_t)}^2\right] \le \sigma^2$.[8] Unlike our results, their neighborhood can be controlled with a smaller stepsize if interpolation does not hold. Thirdly, our results hold for $\eta \le {}^1\!/\!{}_L$ while they require $\eta \le {}^1\!/\!{}_{2L}$. Finally, when $f$ is strongly convex relative to $\psi$ they require $f_\xi$ to be convex while we allow $f_\xi$ to be non-convex.

### 5.3.1 An example of Theorem 1

To demonstrate the differences with previous works we consider a finite-sum example with SPGD and $\psi = {}^1\!/\!{}_2\,\|\cdot\|_2^2$, where each $f_i : \mathbb{R} \to \mathbb{R} : x \mapsto a_i x^2 + b_i x + c_i$ is quadratic, and the constraint is the closed interval $\mathcal{X} = [0, 1]$. As demonstrated in Figure 1, $f$ is the average of both non-convex and strongly convex quadratics, and $f$ is strongly convex relative to $\psi$. Additionally, each $f_i$ is $\max_j 2|a_j|$-smooth relative to $\psi$. More importantly, as depicted in Figure 1 we consider the case where $\sigma_\mathcal{X}^2 = 0$, interpolation relative to $\mathcal{X}$ holds with $f(0) = f_i^*(\mathcal{X})$ for all $i$.

In comparison to Hanzely & Richtarik (2021), their results apply with a neighborhood of convergence equal to $\approx 168$. We note that their neighborhood depends on the choice of stepsize and we report the smallest neighborhood guaranteed by their results by using their perscribed stepsize.

In comparison to Dragomir et al. (2021), their results do not apply for several reasons: SPGD is not included in their analysis (only SGD in the Euclidean case is allowed), $f_i$ is not always convex, and $\nabla f(x_*) \ne 0$. Nevertheless, their variance term in the Euclidean case corresponds to $\mathbb{E}\left[\|\nabla f_i(x_*)\|_2^2\right]$, the expected squared norm at the optimum, which has a value of 520 in this constrained finite-sum example.

## 6 Convergence of mirror SPS

In this section we present our convergence results for SMD with $\mathrm{mSPS}_{\max}$ when $f_i$ are $L_i$-smooth and with varying assumptions. First, we consider the case when $f$ is strongly convex relative to $\psi$, a common assumption when analysing mirror descent under strong convexity Hazan & Kale (2014). Then we present

---

[8]Note that Dragomir et al. (2021) assume $\psi$ to be twice differentiable and strictly convex and $\psi^*$ is the conjugate function of $\psi$. $z_t$ is some point within the line segment between $\nabla\psi(x_t) - 2\eta\nabla f_i(x_*)$ and $\nabla\psi(x_t)$.

rates under convexity and smoothness but without relatively strong convexity. Afterwards, we discuss the results under interpolation and provide examples.

## 6.1 Smooth and strongly convex

With strong convexity of $\psi$ and $f$ being relatively strongly convex with respect to $\psi$ we can show a linear rate of convergence to a neighborhood.

**Theorem 5.** *Assume $f_\xi$ is convex and $L$-smooth almost surely with respect to the norm $||\cdot||$. Furthermore, assume that $f$ is $\mu$-strongly convex relative to $\psi$ over $\mathcal{X}$, where $\psi$ is $\mu_\psi$-strongly convex over $\mathcal{X}$ with respect to the norm $||\cdot||$ and assumption 1 holds. Then SMD with $mSPS_{max}$ and $c \geq \frac{1}{2}$ guarantees*

$$\mathbb{E}\left[B_\psi(x_*; x_{t+1})\right] \leq (1 - \mu\alpha)^t B_\psi(x_*; x_1) + \frac{\eta_b \sigma^2}{\alpha\mu}.$$

*Where $\alpha := \min\{\mu_\psi/2cL, \eta_b\}$.*

Since $\psi$ is strongly convex we get a guarantee on expected distance to the minimum, as $\frac{\mu_\psi}{2}||x_* - x_{t+1}||^2 \leq B_\psi(x_*; x_{t+1})$. Also, if each $f_i$ is a strongly convex function or if it satisfies the Polyak-Łojasiewicz (PL) condition (Assumption 2) then mSPS is upper bounded by equation (10) and equivalent to $mSPS_{max}$ with $\eta_b = \mu_\psi/2c\mu$. Therefore, SMD with mSPS converges by Theorem 5.

A similar result was shown for SGD with $SPS_{max}$ Loizou et al. (2021)[Theorem 3.1]. Indeed, Theorem 5 generalizes their results; by taking $\psi(x) = \frac{1}{2}||x||_2^2$ we recover a result which is true for both SGD and SPGD.

**Corollary 6.** *Assume $f_\xi$ is convex and $L$-smooth with respect to the norm $||\cdot||_2$ almost surely and that $f$ is $\mu$-strongly convex with respect to $||\cdot||_2$ over $\mathcal{X}$. Then SPGD (SGD if $\mathcal{X} = \mathbb{R}^d$) with $SPS_{max}$ guarantees $\mathbb{E}\left[\frac{1}{2}||x_* - x_{t+1}||_2^2\right] \leq (1 - \mu\alpha)^t \frac{1}{2}||x_* - x_1||_2^2 + \frac{\eta_b\sigma^2}{\alpha\mu}$.*

For the case of preconditioned SGD, $\psi(x) = \frac{1}{2}||x||_M^2$ and $\mathcal{X} = \mathbb{R}^d$, we can go further and extend a non-convex result similar to Theorem 3.6 in Loizou et al. (2021). We include the result and proof in section G.3.

In Loizou et al. (2021) constant stepsize results are derived as a special case of $SPS_{max}$, similarly if in $mSPS_{max}$ $\eta_b$ is selected such that $\eta_b \leq \mu_\psi/2cL_{max}$ then $\eta_t$ is a constant and we can derive new constant stepsize results for SMD. However, using Theorem 5 and $mSPS_{max}$ to analyze constant stepsize SMD yields weaker results than Theorem 1. The assumptions made in Theorem 5 are stronger. For example, Theorem 5 requires $\psi$ to be both strongly convex and smooth on $\mathcal{X}$ with respect to a norm which would not be possible if $\psi$ is Legendre over $\mathcal{X}$ and $\mathcal{X}$ is bounded. This limitation, however, does not apply for Theorem 1 and the next result for smooth convex losses since we do not enforce a smoothness condition on $\psi$.

## 6.2 Smooth and convex

Without $f$ being relatively strongly convex we can attain convergence results on the average function value.

**Theorem 7.** *If $f_\xi$ is convex and $L$-smooth with respect to a norm $||\cdot||$ almost surely, assumption 1 holds, and $\psi$ is $\mu_\psi$-strongly convex over $\mathcal{X}$ with respect to $||\cdot||$. Then mirror descent with $mSPS_{max}$ and $c \geq 1$ guarantees*

$$\mathbb{E}\left[f(\bar{x}_t) - f(x_*)\right] \leq \frac{2B_\psi(x_*; x_1)}{\alpha t} + \frac{2\eta_b\sigma^2}{\alpha}.$$

*Where $\alpha := \min\{\mu_\psi/2cL, \eta_b\}$.*

Similarly to Theorem 5 we can derive constant stepsize results, except we require $\eta_b \leq \psi/2L_{max}$ (with $c = 1$), see Section G.2.1 for details. Unlike Theorem 5, however, this result does not require $\psi$ to be smooth over $\mathcal{X}$.

**Comparison with SPS.** Unlike the analysis of SPS our results and stepsize depend on the choice of the mirror map $\psi$. This dependence, as observed historically, is an important motivation for mirror descent,

allowing for tighter bounds and better dependence on the dimension $d$. For example, suppose $\mathcal{X} = \Delta_d$ and $f_\xi$ is $L$ smooth with respect to $||\cdot||_1$ and for simplicity $x_1 = (1/d, \cdots, 1/d)$, $c = 1$, and $\eta_b$ is selected large enough such that $\alpha = \mu_\psi/2cL_{\max}$. Then the bound in Theorem (7) for EG gives $\mathbb{E}[f(\bar{x}_t) - f(x_*)] \leq 4L\log d/t + 4L\eta_b\sigma^2$. Meanwhile, under SPGD the bound is $4dL/t + 4dL\eta_b\sigma^2$, since $f_\xi$ is $\tilde{L}$ smooth with respect to $||\cdot||_2$ if $\tilde{L} = dL$. Note that unlike SPGD the neighborhood of convergence for EG is independent of $d$! Moreover, under interpolation EG converges at a rate that scales logarithmically in $d$, which is otherwise not possible with SGD. Therefore, selecting the appropriate $\psi$ and stepsize allows for better dependence on $d$ with a smaller neighborhood of convergence.

### 6.3 Exact convergence with adaptive stepsizes and interpolation

As a consequence of the previous results, we have several convergence guarantees under interpolation ($\sigma^2 = 0$). In fact, when $\sigma^2 = 0$ the upper bound $\eta_b$ is not needed, the unbounded variant mSPS will enjoy the same convergence rates as mSPS$_{\max}$. Additionally, similar to Section 5, we can attain almost sure convergence results analogous to Corollary 2 and Corollary 4. To the best of our knowledge, all existing results are with constant stepsize (Section 5, Dragomir et al., 2021; Azizan & Hassibi, 2019), or with conditions on the initialization of parameters Azizan et al. (2019). In contrast, with mSPS we have provided exact global convergence guarantees with an adaptive stepsize.

### 6.4 Mirror descent examples

To demonstrate the generality of our results we consider two cases of Theorem 7. We examine the so called $p$-norm algorithms, and preconditioned SGD. Similar results can also be derived with the exponential gradient algorithm and the norm $||\cdot||_1$.

**Corollary 8** (p-norm). *Suppose the assumptions of Theorem 7 hold with $||\cdot|| = ||\cdot||_p$ for $1 < p \leq 2$ and $\psi(x) = \frac{1}{2}||x||_p^2$. Let $q$ be such that $\frac{1}{p} + \frac{1}{q} = 1$. Then SMD with stepsizes $\eta_t = \min\left\{ \frac{(p-1)(f_{\xi_t}(x_t) - f_{\xi_t}^*)}{||\nabla f_{\xi_t}(x_t)||_q^2}, \eta_b \right\}$, guarantees $\mathbb{E}[f(\bar{x}_t) - f(x_*)] \leq \frac{2B_\psi(x_*;x_1)}{\alpha t} + \frac{2\eta_b\sigma^2}{\alpha}$.*

Another interesting case is SGD with preconditioning $x_{t+1} = x_t - \eta M^{-1}\nabla f_i(x_t)$, for some positive definite matrix $M$. In other words, $\psi$ is taken to be $\psi(x) = \frac{1}{2}||x||_M^2$, with $B_\psi(x;y) = \frac{1}{2}||x - y||_M^2$ and $\mathcal{X} = \mathbb{R}^d$.

**Corollary 9** (Preconditioned SGD). *Suppose $\mathcal{X} = \mathbb{R}^d$ and the assumptions of Theorem 7 hold with $||\cdot|| = ||\cdot||_M$, for a positive definite matrix $M$. Then SMD with $\psi(x) = \frac{1}{2}||x||_M^2$ and stepsizes $\eta_t = \min\left\{ \frac{(f_{\xi_t}(x_t) - f_{\xi_t}^*)}{||\nabla f_{\xi_t}(x_t)||_{M^{-1}}^2}, \eta_b \right\}$, guarantees $\mathbb{E}[f(\bar{x}_t) - f(x_*)] \leq \frac{||x_* - x_1||_M^2}{\alpha t} + \frac{2\eta_b\sigma^2}{\alpha}$.*

## 7 Experiments

We test the performance of mSPS on different supervised learning domains and with different instances of SMD. We use mSPS in our convex experiments with $c = 1$. In theory the bounded stepsize mSPS$_{\max}$ is required in absence of interpolation, however, in practice we observe mSPS converges, likely due to the problems being close to interpolation. For our non-convex deep learning experiments we follow Loizou et al. (2021) by selecting $c = 0.2$ and a smoothing procedure to set a moving upper bound for mSPS$_{\max}$.[9] To compare against a constant stepsize we sweep over $\{10^{-5}, 10^{-4}, 10^{-3}, 10^{-2}, 10^{-1}, 1, 10^1, 10^2, 10^3, 10^4, 10^5\}$. Code for our experiments and implementation is available at: `https://github.com/IssamLaradji/mirror-sps`.

We consider 4 series of experiments. First, we consider unconstrained convex problems with mSPS and different $p$-norm algorithms, $\psi(x) = ||x||_p^2$. Second, we evaluate the performance of mSPS with SPGD and positive constraints. Third, we solve a convex problem with a $\ell_1$ constraint using mSPS and the exponentiated gradient algorithm (EG). Finally, Section 7.4 demonstrates that our method shows competitive performance over highly tuned stepsizes for deep learning without any tuning of the hyper-parameters.

---

[9]This technique is a moving upperbound. More precisely we run mSPS$_{\max}$ with an upper bound at time $t$ given by $\eta_b^t = \tau^{b/n}\eta_{t-1}$ where $b$ and $n$ are the batchsize and number of examples respectively, which amounts to $\tau^{b/n} \approx 1$ in our experiments, with $\eta_b^t \approx \eta_{t-1}$.

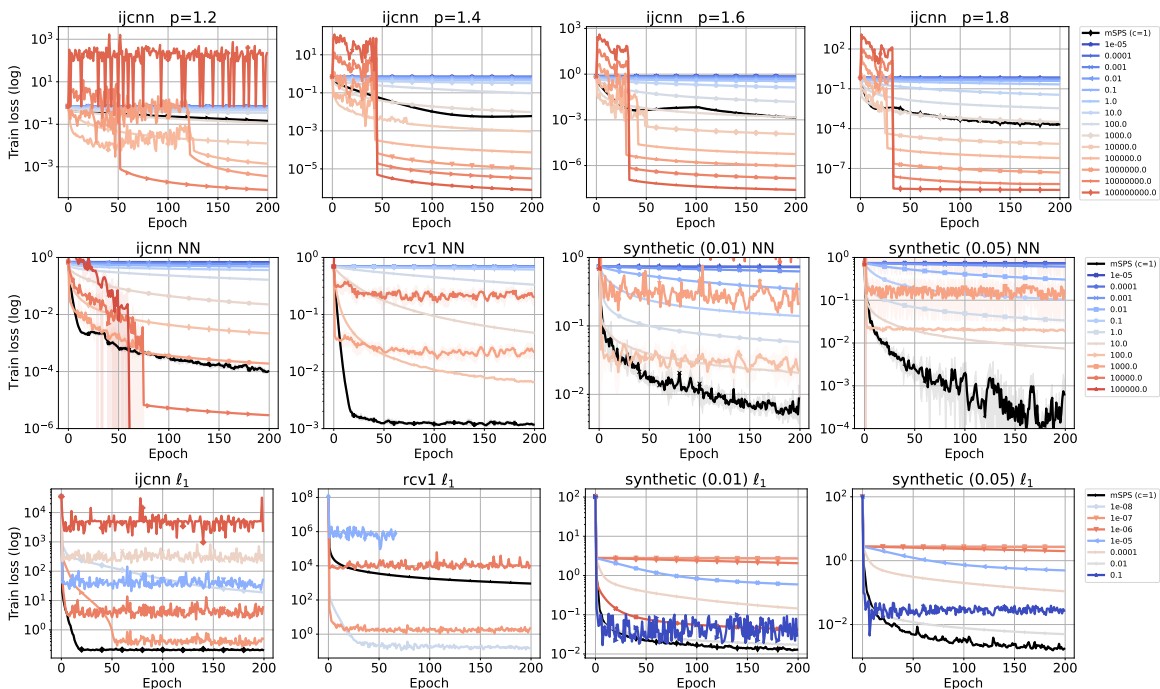

Figure 2: Comparison between mSPS with $c = 1$ and constant stepsizes on convex binary-classification problem with no constraints (row1), with non-negative (NN) constraints (row2), and with $\ell_1$ constraints (row3).

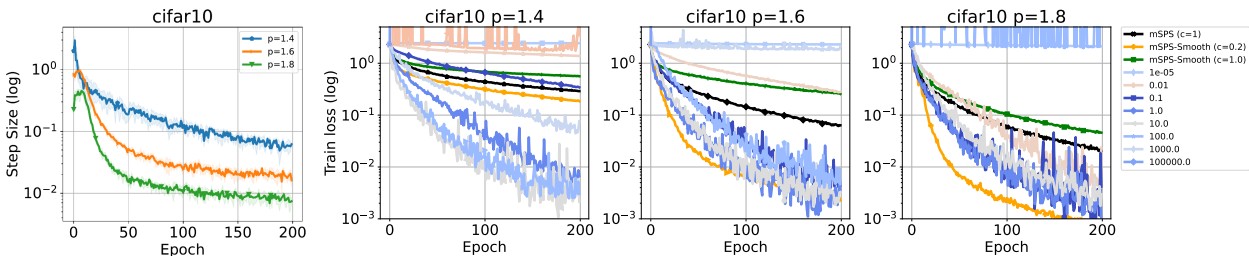

Figure 3: Comparison between mSPS and constant stepsizes on non-convex multiclass classification with deep networks. The leftmost plot shows the stepsize evolution of mSPS with smoothing and $c = 0.2$ for different $p$ values.

## 7.1 Mirror descent across p-norms.

We consider a convex binary-classification problems using radial basis function (RBF) kernels. We experiment on the ijcnn dataset obtained from LIBSVM (Chang & Lin, 2011) which does not satisfy interpolation.[10] ijcnn has 22 dimensions, 39,992 training examples, and 9998 test examples. We selected the kernel bandwidth 0.05 following Vaswani et al. (2019b). For these experiments we compare across $p \in \{1.2, 1.4, 1.6, 1.8\}$ between mSPS and the standard constant stepsize. The first row of Figure 2 shows the training loss for the different optimizers with a softmax loss. We make the following observations: (i) mSPS performs reasonably well across different values of $p$ and outperforms most stepsizes of SMD. (ii) mSPS performs well on ijcnn even though it is not separable in the kernel space (*i.e.* there is no interpolation). This demonstrates some robustness to violations of the interpolation condition and to different values of the $p$. Note that each optimizer was ran with five different random seeds to demonstrate their robustness.

---

[10]In the appendix we include results on the mushroom dataset where interpolation is satisfied.

## 7.2 Projected gradient descent

In this setup we consider optimizing the logistic loss with a non-negative constraint on the parameters. We run our optimizers on two real-datasets ijcnn and rcv1. rcv1 has 47,236 dimensions, 16194 training examples and 4048 test examples. Following Vaswani et al. (2019b) we selected the RBF kernel on rcv1 with bandwidth 0.25.

We also ran the optimizers on two synthetic datasets for binary classification that are linearly separable datasets with margins 0.01 and 0.05 respectively. Linear separability ensures the interpolation condition will hold. For each margin, we generate a dataset with 10k examples with d = 20 features and binary targets.

We observe in the second row of Figure 2 that mSPS outperforms the best tuned constant stepsize in most cases, and in the rest of the cases is competitive. The result underlines the importance of adaptive stepsizes.

## 7.3 Exponentiated gradient

To test the effectiveness of mSPS with EG we consider the datasets in Section 7.2 with logitistic regression where parameters are constrained to the $\ell_1$ ball, $\mathcal{X} = \{x : \|x\|_1 \leq \lambda\}$. To solve this problem with EG, we employ the common trick of reducing an $\ell_1$ ball constraint to a simplex constraint with dimension $(2d-1)$ (Schuurmans & Zinkevich, 2016).

For these experiments we test our optimizers on rcv1 and ijcnn and the two synthetic datasets mentioned in Section 7.2 and report their results in row 3 of Figure 2. Like in the previous experiments, mSPS is significantly faster than most constant stepsizes even though $c$ is kept at 1 and in some cases outperforms the best tuned SMD. Note that the constant stepsizes that don't appear in the plots have diverged.

## 7.4 $p$-norm for optimizing deep networks

For mutliclass-classification with deep networks, we considered the $p$-norm algorithms for the CIFAR10 dataset. CIFAR10 has 10 classes and we used the standard training set consisting of 50k examples and a test set of 10k. As in the kernel experiments, we evaluated the optimizers using the softmax loss for different values of $p$. We used the experimental setup proposed in Loizou et al. (2021) and used a batch-size of 128 for all methods and datasets. We used the standard image-classification architecture ResNet-34 (He et al., 2016). As in the other experiments, each optimizer was run with five different random seeds in the final experiment. The optimizers were run until the performance of most methods saturated; 200 epochs for the models on the CIFAR10 dataset.

From Figure 3, we observe that: (i) mSPS with $c = 0.2$ and smoothing constantly converges to a good solution much faster when compared to most constant stepsizes. (ii) The gap between the performance of mSPS and constant stepsize increases as $p$ decreases suggesting that, like in the convex setting, our method is robust to different values of $p$.

# 8 Conclusions and future work

Stochastic mirror descent (SMD) is a powerful generalization of stochastic projected gradient descent to solve problems without a Euclidean structure. We provide new convergence analysis for SMD with constant step-size in relatively smooth optimization and with the new adaptive stepsizes mSPS, mSPS$_{\max}$, in the smooth case. Consequently, we achieve the first interpolation results for the EG algorithm under interpolation.

A main novelty of our results is the use of the finite optimal objective difference assumption (Loizou et al., 2021) with mirror descent, allowing for convergence without bounded gradient or variance assumptions and achieving exact convergence under interpolation. In relative smooth optimization we refine the finite optimal objective difference assumption to better capture interpolation with constraints and achieve convergence in cases not guaranteed by existing works.

In smooth optimization we experimentally validate mSPS in several supervised learning domains and across various instances of mirror descent. mSPS requires no tuning but is nonetheless competitive or better than

extensively hand-tuned step sizes. This adaptivity is important for tackling different problem domains with different versions of mirror descent.

Beyond the scope of this paper there are several interesting directions for future work. For example, we critically rely on the relative smoothness or smoothness, however, it would be interesting to attain rates of convergence with the finite optimal objective difference assumption without smoothness. Additionally, our convergence result of $\mathrm{mSPS}_{\max}$ in Theorem 5 requires $\psi$ to be smooth over $\mathcal{X}$, an assumption not required for our constant stepsize results in Section 5, it would be interesting to unify the results by developing a variant of $\mathrm{mSPS}_{\max}$ for the more general relatively smooth problem.

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

## A  Appendix

The appendices include omitted proofs, other results, and additional experiments. The material is organized as follows: equivalent definitions of relative smoothness are given in Section B; standard mirror descent results are presented in Section C; non-smooth analysis of mirror descent with the mirror Polyak stepsize is given in Section D; the lower bound proof of mSPS is included in Section E; the proofs for the results in Section 5 are presented in Section F, including Theorem 1 and Theorem 3; proofs for Section 6 are given in Section G, including a non-convex result for preconditioned SGD in Section G.3; experiment details are given in Section H.

## B  Relative Smoothness

Relative smoothness is a generalization of smoothness in first order optimization that includes non-Lipschitz gradients. A function is $L$-smooth with respect to a norm $||\cdot||$ if

$$||\nabla f(x) - \nabla f(y)||_* \leq L \, ||x - y|| \, .$$

With a Lipschitz gradient the error in the first order approximation of $f$ grows at most quadratically

$$f(x) - (f(y) - \langle \nabla f(y), x - y \rangle) = B_f(x; y) \leq \frac{L}{2} \, ||x - y||^2 \, .$$

Relative smoothness replaces the quadratic upper bound with a divergence relative to a convex function $\psi$,

$$B_f(x; y) \leq L B_\psi(x; y).$$

If $\psi$ is strongly convex then smoothness with respect to a norm implies relative smoothness with respect to $\psi$. However, a relative smooth function may not admit a Lipchitz gradient, as was first remarked by Birnbaum et al. (2011) and later rediscovered by Lu et al. (2018) and Bauschke et al. (2017). Similar to traditional smoothness, equivalent definitions of relative smoothness exist in the literature. For example, Lu et al. (2018) prove that the following conditions are equivalent:

1. $f$ is $L$ smooth relative to $\psi$

2. $L\psi - f$ is convex

3. under twice differentiability $\nabla^2 f \preceq L\nabla^2\psi$

4. $\langle \nabla f(x) - \nabla f(y), x - y \rangle \leq L\langle \nabla\psi(x) - \nabla\psi(y), x - y \rangle.$

Similar conditions can also be stated for relative strong convexity.

Defining curvature and smoothness relative to a function $\psi$ allows for a wider application of first order methods via mirror descent. Recently, such assumptions have been used to establish both new results and algorithms in machine learning. In reinforcement learning, Vaswani et al. (2022) use relative smoothness and mirror descent to provide a new perspective on policy optimization in reinforcement learning. In algorithmic game theory, Sokota et al. (2023) use relative strong convexity to establish fast convergence to quantal-response equilibria in extensive-form games.

## C   Mirror descent lemmas

**Lemma 3** (Three Point Property (Chen & Teboulle, 1993)). *Let $B_\psi$ be the Bregman divergence with respect to $\psi : \mathcal{D} \to \mathbb{R}$. Then for any three points $x, y \in \operatorname{int} \mathcal{D}$ , and $z \in \mathcal{D}$, the following holds*

$$B_\psi(z; x) + B_\psi(x; y) - B_\psi(z; y) = \langle \nabla\psi(y) - \nabla\psi(x), z - x \rangle.$$

### C.1   Proof of Lemma 1

**Lemma 1.** *Let $B_\psi$ be the Bregman divergence with respect to a convex function $\psi : \mathcal{D} \to \mathbb{R}$ and assume assumption 1 holds. Let $x_{t+1} = \arg\min_{x \in \mathcal{X}} \langle g_t, x \rangle + \frac{1}{\eta_t} B_\psi(x; x_t)$. Then for any $x_* \in \mathcal{X}$*

$$B_\psi(x_*; x_{t+1}) \leq B_\psi(x_*; x_t) - \eta_t\langle g_t, x_t - x_* \rangle - B_\psi(x_{t+1}; x_t) + \eta_t\langle g_t, x_t - x_{t+1} \rangle. \tag{14}$$

*Furthermore if $\psi$ is $\mu_\psi$ strongly convex over $\mathcal{X}$ then*

$$B_\psi(x_*; x_{t+1}) \leq B_\psi(x_*; x_t) - \eta_t\langle g_t, x_t - x_* \rangle + \frac{\eta_t^2}{2\mu_\psi} \|g_t\|_*^2. \tag{15}$$

*Proof.* The proof follows closely to the one presented in Orabona (2019)[Lemma 6.7]. First observe that $x_{t+1}$ statisfies the first order optimality condition

$$\langle \eta_t g_t + \nabla\psi(x_{t+1}) - \nabla\psi(x_t), x_* - x_{t+1} \rangle \geq 0,$$

since $\nabla_x B_\psi(x; x_t) = \nabla\psi(x) - \nabla\psi(x_t)$.

We start by examining the inner product $\langle \eta_t g_t, x_t - x_* \rangle$ and adding subtracting quantities to make the first order optimality condition appear.

$$
\begin{aligned}
\langle \eta_t g_t, x_t - x_* \rangle &= \langle \eta_t g_t + \nabla\psi(x_{t+1}) - \nabla\psi(x_t), x_{t+1} - x_* \rangle + \langle \nabla\psi(x_{t+1}) - \nabla\psi(x_t), x_* - x_{t+1} \rangle + \langle \eta_t g_t, x_t - x_{t+1} \rangle \\
&\leq \langle \nabla\psi(x_{t+1}) - \nabla\psi(x_t), x_* - x_{t+1} \rangle + \langle \eta_t g_t, x_t - x_{t+1} \rangle \text{(first order optimality)} \\
&= B_\psi(x_*; x_t) - B_\psi(x_*; x_{t+1}) - B_\psi(x_{t+1}; x_t) + \langle \eta_t g_t, x_t - x_{t+1} \rangle \text{ (three point property)}.
\end{aligned}
$$

Rearranging gives the first result. Note at this point we only require $\psi$ to be convex and $\psi$ to be differentiable at $x_t$ and $x_{t+1}$, which is guaranteed by assumption 1. To obtain the second result, observe

$$
\begin{aligned}
\langle \eta_t g_t, x_t - x_* \rangle &\leq B_\psi(x_*; x_t) - B_\psi(x_*; x_{t+1}) - B_\psi(x_{t+1}; x_t) + \langle \eta_t g_t, x_t - x_{t+1} \rangle \text{ (from above)} \\
&\leq B_\psi(x_*; x_t) - B_\psi(x_*; x_{t+1}) - \frac{\mu_\psi}{2} \|x_{t+1} - x_t\|^2 + \langle \eta_t g_t, x_t - x_{t+1} \rangle \text{ (strong convexity)} \\
&\leq B_\psi(x_*; x_t) - B_\psi(x_*; x_{t+1}) + \frac{\eta_t^2}{2\mu_\psi} \|g_t\|_*^2 \text{ (Fenchel-Young inequality)}.
\end{aligned}
$$

Rearranging gives the second result. $\qquad\square$

# D  Non-smooth analysis of mirror SPS for Lipschitz functions

As we have already mentioned in the main paper, the Polyak step-size is used extensively in the literature of projected subgradient descent for solving non-smooth optimization problems. However to the best of our knowledge there is no efficient generalization of this step-size for the more general mirror descent update.

**Theorem 10** (Non-smooth deterministic). *Assume $f$ is convex with bounded subgradients, $||\partial f(x_t)||_* \leq G$. Let $\psi$ be $\mu_\psi$ strongly convex with respect to the norm $||\cdot||$, and assume that Assumption 1 holds. Then mirror descent with stepsize $\eta_t = \frac{\mu_\psi(f(x_t) - f(x_*))}{||\partial f(x_t)||_*^2}$ satisfies,*

$$f(\bar{x}_t) - f(x_*) \leq G\sqrt{\frac{\frac{2}{\mu_\psi} B_\psi(x_*; x_1)}{t}},$$

*where $\bar{x}_t = \frac{1}{t}\sum_{s=1}^{t} x_s$. The same result holds for the best iterate $f(x_t^*) = \min_s\{f(x_s)\}_{1 \leq s \leq t}$. Moreover, we have $\lim_{t \to \infty} f(x_t) = f(x_*)$.*

*Proof.* Let $g_t$ be a subgradient of $f$ at $x_t$ used to compute $\eta_t$. Then by Lemma 1 we have

$$B_\psi(x_*; x_{t+1}) \leq B_\psi(x_*; x_t) - \eta_t\langle g_t, x_t - x_*\rangle + \frac{\eta_t^2}{2\mu_\psi}||g_t||_*^2$$

$$\leq B_\psi(x_*; x_t) - \eta_t(f(x_t) - f(x_*)) + \frac{\eta_t^2}{2\mu_\psi}||g_t||_*^2 \text{ (by convexity)}$$

$$= B_\psi(x_*; x_t) - \frac{\mu_\psi(f(x_t) - f(x_*))^2}{||g_t||_*^2} + \frac{\mu_\psi(f(x_t) - f(x_*))^2}{2||g_t||_*^2} \text{ (by definition of } \eta_t)$$

$$= B_\psi(x_*; x_t) - \frac{\mu_\psi(f(x_t) - f(x_*))^2}{2||g_t||_*^2}.$$

Rearranging and summing across time we have

$$\sum_{s=1}^{t} \frac{\mu_\psi(f(x_s) - f(x_*))^2}{2||g_s||_*^2} \leq B_\psi(x_*; x_1) - B_\psi(x_*; x_{t+1}) \leq B_\psi(x_*; x_1). \tag{16}$$

Applying the upper bound $||g_s||_* \leq G$ and taking the square root gives,

$$\sqrt{\sum_{s=1}^{t}(f(x_s) - f(x_*))^2} \leq G\sqrt{\frac{2B_\psi(x_*; x_1)}{\mu_\psi}}.$$

The result then follows by the convexity of $f$ and concavity of the square root function,

$$f(\bar{x}_t) - f(x_*) \leq \frac{1}{t}\sum_{s=1}^{t}(f(x_s) - f(x_*)) = \frac{1}{t}\sum_{s=1}^{t}\sqrt{(f(x_s) - f(x_*))^2} \leq \sqrt{\frac{1}{t}\sum_{s=1}^{t}(f(x_s) - f(x_*))^2}$$

$$\leq G\sqrt{\frac{2B_\psi(x_*; x_1)}{t\mu_\psi}}.$$

To obtain the best iterate result notice that $f(x_t^*) - f(x_*)) \leq \frac{1}{t}\sum_{s=1}^{t}(f(x_s) - f(x_*))$.

To attain the limiting result observe that 16 implies

$$\sum_{s=1}^{\infty} \mu_\psi(f(x_s) - f(x_*))^2 \leq G^2 B_\psi(x_*; x_1) < +\infty.$$

Giving the result $\lim_{t \to \infty} f(x_t) = f(x_*)$.  □

### D.1 Last-iterate convergence to a solution

Under the same assumptions as Theorem 10 we have that mirror descent converges to a point. First we provide a useful lemma applicable to mirror descent with a strongly convex mirror map $\psi$.

**Lemma 4.** *Suppose $\psi$ is strongly convex, then if the sequece $\{x_t\}_{t \geq 1}$ is Bregman monotone with respect to a set $\mathcal{S}$, that is for any $x \in \mathcal{S}$ we have*

$$B_\psi(x; x_{t+1}) \leq B_\psi(x; x_t),$$

*then $x_t \to x_* \in S$ if and only if all the sequential cluster points of $\{x_t\}_{t \geq 1}$ are contained in $\mathcal{S}$.*

*Proof.* If the sequence is $\{x_t\}_{t \geq 1}$ is Bregman monotone then by strong convexity

$$\frac{\mu_\psi}{2} \|x - x_{t+1}\|^2 \leq B_\psi(x; x_1),$$

hence the sequence is bounded. Therefore, the sequence has a limit point $x_l$ such that there exists a subsequence $x_{b_t} \to x_l$. Assume $x_l \in \mathcal{S}$ and consider the sequence $\{y_t = B_\psi(x_l; x_t)\}_{t \geq 1}$. Since $y_t$ is monotonically decreasing and bounded below $y_t \to L$ for some $L \in \mathbb{R}$. However, the subsequence $\{B_\psi(x_l; x_{b_t})\}$ converges to zero, therefore we have that $\lim_{t \to \infty} B_\psi(x_l; x_t) = 0$, implying that $\lim_{t \to \infty} x_t = x_l$ by strong convexity of $\psi$. $\qquad\square$

**Corollary 11.** *Under the same assumptions as Theorem 10, mirror descent converges to a solution,*

$$\lim_{t \to \infty} x_t = x_*,$$

*for some $x_* \in \mathcal{X}_*$.*

*Proof.* From Theorem 10 we have the following inequality,

$$B_\psi(x_*; x_{t+1}) \leq B_\psi(x_*; x_t) - \frac{\mu_\psi \left(f(x_t) - f(x_*)\right)^2}{2 \|g_t\|_*^2}.$$

Therefore by Lemma 4 it remains to show that all limit points of $x_t$ are contained within $\mathcal{X}_*$.

By Theorem 10 we have that $f(x_t) \to f(x_*)$. For any limit point $x_l$ we have a subsequence $x_{b_t}$ such that $x_{b_t} \to x_l$ and by continuity of $f$, $x_l$ must be a solution,

$$f(x_*) = \lim_{t \to \infty} f(x_t) = \lim_{t \to \infty} f(x_{b_t}) = f(\lim_{t \to \infty} x_{b_t}) = f(x_l).$$

The result then follows by Lemma 4. $\qquad\square$

## E   Proof of mSPS lower bound in section 4

The lower bound of mSPS (10) when $f_\xi$ is L smooth, restated below, is vital to our analysis,

$$\frac{\mu_\psi}{2cL} \leq \eta_t = \frac{\mu_\psi(f_\xi(x_t) - f_\xi^*)}{c \|\nabla f(x_t)\|_*^2}.$$

Notice the above inequality is equivalent to

$$\frac{1}{2L} \leq \frac{(f_\xi(x_t) - f_\xi^*)}{\|\nabla f(x_t)\|_*^2}.$$

The first inequality is attained by multiplying both sides by $\mu_\psi/c$. We provide a detailed proof below.

**Lemma 5.** *If $f : \mathbb{R}^n \to \mathbb{R}$ is $L$-smooth with respect to a norm $||\cdot||$ then*

$$\frac{||\nabla f(x)||_*^2}{2L} \leq f(x) - \inf_{y \in \mathbb{R}^n} f(y).$$

*Rearranging and defining $f^* = \inf_{y \in \mathbb{R}^n} f(y)$ gives*

$$\frac{1}{2L} \leq \frac{f(x) - f^*}{||\nabla f(x)||_*^2}.$$

*Proof.* Since $f$ is L-smooth we have

$$f(y) \leq f(x) + \langle \nabla f(x), y - x \rangle + \frac{L}{2} ||x - y||^2 \quad \forall x, y \in \mathbb{R}^n.$$

Therefore we have the following upper bound on $\inf_y f(y)$.

$$
\begin{aligned}
\inf_y f(y) &\leq \min_y \left\{ f(x) + \langle \nabla f(x), y - x \rangle + \frac{L}{2} ||x - y||^2 \right\} \\
&= \min_{r \geq 0, ||z|| \leq 1} \left\{ f(x) + r\langle \nabla f(x), z \rangle + \frac{L}{2} r^2 ||z||^2 \right\} \\
&\leq \min_{r \geq 0, ||z|| \leq 1} \left\{ f(x) + r\langle \nabla f(x), z \rangle + \frac{L}{2} r^2 \right\} \\
&= f(x) + \min_{r \geq 0} \left\{ \min_{||z|| \leq 1} \left\{ r\langle \nabla f(x), z \rangle \right\} + \frac{L}{2} r^2 \right\} \\
&= f(x) + \min_{r \geq 0} \left\{ -r \max_{||z|| \leq 1} \left\{ \langle \nabla f(x), -z \rangle \right\} + \frac{L}{2} r^2 \right\} \\
&= f(x) + \min_{r \geq 0} \left\{ -r ||\nabla f(x)||_* + \frac{L}{2} r^2 \right\} \text{ by the definition of } ||\cdot||_* \\
&\stackrel{(r = ||\nabla f(x)||_*/L)}{=} f(x) - \frac{||\nabla f(x)||_*^2}{L} + \frac{||\nabla f(x)||_*^2}{2L}
\end{aligned}
$$

Simplifying and rearranging gives the result. $\qquad\square$

# F   Proofs for section 5

In this section we provide proofs of our main results in the relative smooth setting. For convenience we denote the expectation conditional upon $(\xi_1, \xi_2, \cdots, \xi_t)$ as $\mathbb{E}_t[\cdot]$. All statements hold almost surely.

## F.1   Proof of Lemma 2

**Lemma 2.** *Suppose $f$ is $L$ smooth relative to $\psi$. Then if $\eta \leq \frac{1}{L}$ we have*

$$-B_\psi(x_{t+1}; x_t) + \eta \langle \nabla f(x_t), x_t - x_{t+1} \rangle \leq \eta(f(x_t) - f(x_{t+1})).$$

*Proof.* Since $f$ is $L$ smooth relative to $\psi$ it is also $\frac{1}{\eta}$ smooth relative to $\psi$ (because $L \leq \frac{1}{\eta}$ and $\psi$ is convex). Therefore,

$$B_f(x_{t+1}; x_t) \leq \frac{1}{\eta} B_\psi(x_{t+1}; x_t)$$
$$\implies -B_\psi(x_{t+1}; x_t) + \eta B_f(x_{t+1}; x_t) \leq 0.$$

Now we examine the inner product $\eta\langle\nabla f(x_t), x_t - x_{t+1}\rangle$,

$$\eta\langle\nabla f(x_t), x_t - x_{t+1}\rangle = \eta\left(f(x_{t+1}) - f(x_t) - \langle\nabla f(x_t), x_{t+1} - x_t\rangle + f(x_t) - f(x_{t+1})\right)$$
$$= \eta\left(B_f(x_{t+1}; x_t) + f(x_t) - f(x_{t+1})\right).$$

Therefore, we have the following

$$-B_\psi(x_{t+1}; x_t) + \eta\langle\nabla f(x_t), x_t - x_{t+1}\rangle = -B_\psi(x_{t+1}; x_t) + \eta B_f(x_{t+1}; x_t) + \eta(f(x_t) - f(x_{t+1})$$
$$\leq \eta(f(x_t) - f(x_{t+1})).$$

$\square$

## F.2 Proof of Theorem 1

**Theorem 1.** *Assume $\psi$ satisfies assumption 1. Furthermore assume $f$ to be $\mu$-strongly convex relative to $\psi$ over $\mathcal{X}$, and $f_\xi$ to be $L$-smooth relative to $\psi$ over $\mathcal{X}$ almost surely. Then SMD with stepsize $\eta \leq \frac{1}{L}$ guarantees*

$$\mathbb{E}\left[B_\psi(x_*; x_{t+1})\right] \leq (1 - \mu\eta)^t B_\psi(x_*; x_1) + \frac{\sigma_{\mathcal{X}}^2}{\mu}.$$

*Proof.* From Lemma 1 (before applying strong convexity but assuming convexity of $\psi$) we have

$$B_\psi(x_*; x_{t+1}) \leq B_\psi(x_*; x_t) - \eta\langle\nabla f_{\xi_t}(x_t), x_t - x_*\rangle - B_\psi(x_{t+1}; x_t) + \eta\langle\nabla f_{\xi_t}(x_t), x_t - x_{t+1}\rangle$$
$$\leq B_\psi(x_*; x_t) - \eta\langle\nabla f_{\xi_t}(x_t), x_t - x_*\rangle + \eta(f_{\xi_t}(x_t) - f_{\xi_t}(x_{t+1})) \text{ (by Lemma 2)}$$
$$\leq B_\psi(x_*; x_t) - \eta\langle\nabla f_{\xi_t}(x_t), x_t - x_*\rangle + \eta(f_{\xi_t}(x_t) - f_{\xi_t}^*(\mathcal{X})) \text{ (by definition of } f_{\xi_t}^*(\mathcal{X}))$$
$$= B_\psi(x_*; x_t) - \eta\langle\nabla f_{\xi_t}(x_t), x_t - x_*\rangle + \eta(f_{\xi_t}(x_t) - f_{\xi_t}(x_*)) + \eta(f_{\xi_t}(x_*) - f_{\xi_t}^*(\mathcal{X})).$$

By taking an expectation conditioning on $(\xi_1, \cdots, \xi_t)$ we obtain,

$$\mathbb{E}_t\left[B_\psi(x_*; x_{t+1})\right] \leq B_\psi(x_*; x_t) - \eta\langle\nabla f(x_t), x_t - x_*\rangle + \eta(f(x_t) - f(x_*)) + \eta(f(x_*) - \mathbb{E}_t\left[f_{\xi_t}^*(\mathcal{X})\right])$$
$$= B_\psi(x_*; x_t) - \eta\underbrace{(f(x_*) - f(x_t) - \langle\nabla f(x_t), x_* - x_t\rangle)}_{B_f(x_*; x_t)} + \eta(f(x_*) - \mathbb{E}_t\left[f_{\xi_t}^*(\mathcal{X})\right]) \qquad (17)$$
$$\leq B_\psi(x_*; x_t)(1 - \mu\eta) + \eta(f(x_*) - \mathbb{E}_t\left[f_{\xi_t}^*(\mathcal{X})\right]) \text{ (by relative strongly convexity of } f).$$

Now by the tower property of expectations and applying the definition of $\sigma_{\mathcal{X}}^2$,

$$\mathbb{E}\left[B_\psi(x_*; x_{t+1})\right] \leq \mathbb{E}\left[B_\psi(x_*; x_t)\right](1 - \mu\eta) + \eta\sigma_{\mathcal{X}}^2.$$

Iterating the inequality gives,

$$\mathbb{E}\left[B_\psi(x_*; x_{t+1})\right] \leq B_\psi(x_*; x_1)(1 - \mu\eta)^t + \sum_{s=0}^{t-1}\eta\sigma_{\mathcal{X}}^2(1 - \mu\eta)^s$$

$$\leq B_\psi(x_*; x_1)(1 - \mu\eta)^t + \frac{\sigma_{\mathcal{X}}^2}{\mu}.$$

Where the last inequality follows by $\sum_{s=0}^{t-1}(1 - \mu\eta)^s \leq \sum_{s=0}^{\infty}(1 - \mu\eta)^s = \frac{1}{\mu\eta}$. $\square$

**Corollary 2.** *Under the same assumptions at Theorem 1, if $\sigma_{\mathcal{X}}^2 = 0$ then $B_\psi(x_*; x_{t+1}) \to 0$ almost surely.*

*Proof.* By Theorem 1 the following inequality holds:

$$\mathbb{E}_t\left[B_\psi(x_*; x_{t+1})\right] \leq B_\psi(x_*; x_t)(1 - \mu\eta). \qquad (18)$$

The result then follows by Franci & Grammatico (2022)[Lemma 4.7]. $\square$

### F.3 Proof of Theorem 3

**Theorem 3.** *Assume $\psi$ satisfies assumption 1. Furthermore assume $f_\xi$ to be L-smooth relative to $\psi$ over $\mathcal{X}$ almost surely. Then SMD with stepsize $\eta \leq \frac{1}{L}$ guarantees*

$$\mathbb{E}\left[\frac{1}{t}\sum_{s=1}^{t} B_f(x_*; x_s)\right] \leq \frac{B_\psi(x_*; x_1)}{\eta t} + \sigma_{\mathcal{X}}^2.$$

*Proof.* Note that in the proof of Theorem 1 relative strong convexity is not used to attain the inequality (17). Therefore we have,

$$\mathbb{E}_t\left[B_\psi(x_*; x_{t+1})\right] \leq B_\psi(x_*; x_t) - \eta B_f(x_*; x_t) + \eta(f(x_*) - \mathbb{E}_t\left[f_{\xi_t}^*(\mathcal{X})\right]).$$

After applying the tower property, definition of $\sigma^2$, and rearranging, we have

$$\eta \mathbb{E}\left[B_f(x_*; x_t)\right] \leq \mathbb{E}\left[B_\psi(x_*; x_t)\right] - \mathbb{E}\left[B_\psi(x_*; x_{t+1})\right] + \eta\sigma_{\mathcal{X}}^2.$$

Summing across time and dividing by $\eta t$ gives the result. $\qquad \square$

**Corollary 4.** *Under the assumptions of Theorem 3, if $f$ is convex and $\sigma_{\mathcal{X}}^2 = 0$ then $B_f(x_*; x_t) \to 0$ almost surely.*

*Proof.* Under interpolation $f(x_*) - \mathbb{E}_t\left[f_{\xi_t}^*(\mathcal{X})\right] = 0$. From Theorem 3 the following inequality holds:

$$\mathbb{E}_t\left[B_\psi(x_*; x_{t+1})\right] \leq B_\psi(x_*; x_t) - \eta B_f(x_*; x_t). \tag{19}$$

Since $f$ is convex $B_f(x_*; x_t) \geq 0$, therefore, by the Robbins-Siegmun Lemma (*e.g.* (Franci & Grammatico, 2022)[Lemma 4.1]) $B_f(x_*; x_t) \to 0$ almost surely.

$$\square$$

**Proposition 1.** *Let $f(x) = \frac{1}{2}\left(\langle g, x\rangle - b\right)^2$ then $B_f(x; y) = B_f(y; x)$.*

*Proof.* Note that $\nabla f(x) = \left(\langle g, x\rangle - b\right)g$. Therefore,

$$B_f(x; y) = \frac{1}{2}\left(\langle g, x\rangle - b\right)^2 - \frac{1}{2}\left(\langle g, y\rangle - b\right)^2 - \left(\langle g, y\rangle - b\right)\langle g, x - y\rangle \tag{20}$$

$$= \frac{1}{2}(\langle g, x\rangle)^2 - \langle g, x\rangle b + b^2 - \frac{1}{2}(\langle g, y\rangle)^2 + \langle g, y\rangle b - b^2 - \left(\langle g, y\rangle - b\right)\langle g, x - y\rangle \tag{21}$$

$$= \frac{1}{2}(\langle g, x\rangle)^2 - \langle g, x\rangle b - \frac{1}{2}(\langle g, y\rangle)^2 + \langle g, y\rangle b - \left(\langle g, y\rangle - b\right)\langle g, x - y\rangle \tag{22}$$

$$= \frac{1}{2}(\langle g, x\rangle)^2 - \langle g, x\rangle b - \frac{1}{2}(\langle g, y\rangle)^2 + \langle g, y\rangle b - \langle g, y\rangle\langle g, x\rangle + b\langle g, x\rangle + (\langle g, y\rangle)^2 - b\langle g, y\rangle \tag{23}$$

$$= \frac{1}{2}(\langle g, x\rangle)^2 + \frac{1}{2}(\langle g, y\rangle)^2 - \langle g, y\rangle\langle g, x\rangle. \tag{24}$$

It follows that $B_f$ is symmetric. $\qquad \square$

## G   Proofs for section 6

In this section we provide proofs of our main results in the smooth setting. For convenience we denote the expectation conditional upon $(\xi_1, \xi_2, \cdots, \xi_t)$ as $\mathbb{E}_t\left[\cdot\right]$. All statements hold almost surely.

Notice that by definition of $\text{mSPS}_{\text{max}}$ we have the following upper bound

$$\eta_t \leq \frac{\mu_\psi(f_{\xi_t}(x_t) - f_{\xi_t}^*)}{c\left\|\nabla f_{\xi_t}(x_t)\right\|_*^2}.$$

Muliplying both sides of the inequality with $\eta_t ||\nabla f_{\xi_t}(x_t)||_*^2 / \mu_\psi$ gives the following useful inequality,

$$\frac{\eta_t^2 ||\nabla f_{\xi_t}(x_t)||_*^2}{\mu_\psi} \leq \frac{\eta_t (f_{\xi_t}(x_t) - f_{\xi_t}^*)}{c}. \tag{25}$$

The inequality holds with equality for mSPS.

### G.1 Proof of Theorem 5

**Theorem 5.** *Assume $f_\xi$ is convex and $L$-smooth almost surely with respect to the norm $||\cdot||$. Furthermore, assume that $f$ is $\mu$-strongly convex relative to $\psi$ over $\mathcal{X}$, where $\psi$ is $\mu_\psi$-strongly convex over $\mathcal{X}$ with respect to the norm $||\cdot||$ and assumption 1 holds. Then SMD with $mSPS_{max}$ and $c \geq \frac{1}{2}$ guarantees*

$$\mathbb{E}\left[B_\psi(x_*; x_{t+1})\right] \leq (1 - \mu\alpha)^t B_\psi(x_*; x_1) + \frac{\eta_b \sigma^2}{\alpha\mu}.$$

*Where $\alpha := \min\{\mu_\psi/2cL, \eta_b\}$.*

*Proof.*

$$B_\psi(x_*; x_{t+1}) \leq B_\psi(x_*; x_t) - \eta_t \langle \nabla f_{\xi_t}(x_t), x_t - x_* \rangle + \frac{\eta_t^2}{2\mu_\psi} ||\nabla f_{\xi_t}(x_t)||_*^2$$

$$\overset{(25)}{\leq} B_\psi(x_*; x_t) - \eta_t \langle \nabla f_{\xi_t}(x_t), x_t - x_* \rangle + \eta_t \frac{(f_{\xi_t}(x_t) - f_{\xi_t}^*)}{2c}$$

$$\overset{(c \geq 1/2)}{\leq} B_\psi(x_*; x_t) - \eta_t \langle \nabla f_{\xi_t}(x_t), x_t - x_* \rangle + \eta_t (f_{\xi_t}(x_t) - f_{\xi_t}^*)$$

$$= B_\psi(x_*; x_t) - \eta_t \langle \nabla f_{\xi_t}(x_t), x_t - x_* \rangle + \eta_t (f_{\xi_t}(x_t) - f_{\xi_t}(x_*) + f_{\xi_t}(x_*) - f_{\xi_t}^*)$$

$$= B_\psi(x_*; x_t) - \underbrace{\eta_t \left( f_{\xi_t}(x_*) - f_{\xi_t}(x_t) - \langle \nabla f_{\xi_t}(x_t), x_* - x_t \rangle \right)}_{\geq 0} + \eta_t (f_{\xi_t}(x_*) - f_{\xi_t}^*)$$

$$\overset{(10)}{\leq} B_\psi(x_*; x_t) - \min\left\{\frac{\mu_\psi}{2cL}, \eta_b\right\} (f_{\xi_t}(x_*) - f_{\xi_t}(x_t) - \langle \nabla f_{\xi_t}(x_t), x_* - x_t \rangle) + \eta_b (f_{\xi_t}(x_*) - f_{\xi_t}^*)$$

Taking an expectation over $i$ condition on $x_t$ gives

$$\mathbb{E}_t\left[B_\psi(x_*; x_{t+1})\right] \leq B_\psi(x_*; x_t) - \min\left\{\frac{\mu_\psi}{2cL}, \eta_b\right\} (f(x_*) - f(x_t) - \langle \nabla f(x_t), x_* - x_t \rangle) + \eta_b \mathbb{E}_t\left[(f_{\xi_t}(x_*) - f_{\xi_t}^*)\right]$$

$$\leq B_\psi(x_*; x_t) \left(1 - \mu \min\left\{\frac{\mu_\psi}{2cL_{\max}}, \eta_b\right\}\right) + \eta_b \mathbb{E}_t\left[(f_{\xi_t}(x_*) - f_{\xi_t}^*)\right] \text{ (by relative strong convexity of } f)$$

$$= B_\psi(x_*; x_t) (1 - \mu\alpha) + \eta_b \mathbb{E}_t\left[(f_{\xi_t}(x_*) - f_{\xi_t}^*)\right].$$

Now by the tower property of expectations and applying the definition of $\sigma^2$,

$$\mathbb{E}\left[B_\psi(x_*; x_{t+1})\right] \leq \mathbb{E}\left[B_\psi(x_*; x_t)\right] (1 - \mu\alpha) + \eta_b \sigma^2.$$

Iterating the inequality gives,

$$\mathbb{E}\left[B_\psi(x_*; x_{t+1})\right] \leq B_\psi(x_*; x_1)(1 - \mu\alpha)^t + \sum_{s=0}^{t-1} \eta_b \sigma^2 (1 - \mu\alpha)^s$$

$$\leq B_\psi(x_*; x_1)(1 - \mu\alpha)^t + \frac{\eta_b \sigma^2}{\alpha\mu}.$$

Where the last inequality follows by $\sum_{s=0}^{t-1}(1 - \mu\alpha)^s \leq \sum_{s=0}^{\infty}(1 - \mu\alpha)^s = 1/\mu\alpha$. $\qquad\square$

## G.2 Proof of Theorem 7

**Theorem 7.** *If $f_\xi$ is convex and $L$-smooth with respect to a norm $||\cdot||$ almost surely, assumption 1 holds, and $\psi$ is $\mu_\psi$-strongly convex over $\mathcal{X}$ with respect to $||\cdot||$. Then mirror descent with $mSPS_{max}$ and $c \geq 1$ guarantees*

$$\mathbb{E}\left[f(\bar{x}_t) - f(x_*)\right] \leq \frac{2B_\psi(x_*; x_1)}{\alpha t} + \frac{2\eta_b \sigma^2}{\alpha}.$$

*Where $\alpha \coloneqq \min\{\mu_\psi/2cL, \eta_b\}$.*

*Proof.* We begin with Lemma 1,

$$
\begin{aligned}
B_\psi(x_*; x_{t+1}) &\leq B_\psi(x_*; x_t) - \eta_t \langle \nabla f_{\xi_t}(x_y), x_t - x_* \rangle + \frac{\eta_t^2}{2\mu_\psi} ||\nabla f_{\xi_t}(x_t)||_*^2 \\
&\leq B_\psi(x_*; x_t) - \eta_t \left(f_{\xi_t}(x_t) - f_{\xi_t}(x_*)\right) + \frac{\eta_t^2}{2\mu_\psi} ||\nabla f_{\xi_t}(x_t)||_*^2 \text{ by convexity} \\
&\stackrel{(25)}{\leq} B_\psi(x_*; x_t) - \eta_t \left(f_{\xi_t}(x_t) - f_{\xi_t}(x_*)\right) + \frac{\eta_t (f_{\xi_t}(x_t) - f_{\xi_t}^*)}{2c} \\
&\stackrel{(c \geq 1)}{\leq} B_\psi(x_*; x_t) - \eta_t \left(f_{\xi_t}(x_t) - f_{\xi_t}(x_*)\right) + \frac{\eta_t (f_{\xi_t}(x_t) - f_{\xi_t}^*)}{2} \\
&= B_\psi(x_*; x_t) - \eta_t \left(f_{\xi_t}(x_t) - f_{\xi_t}^* + f_{\xi_t}^* - f_{\xi_t}(x_*)\right) + \frac{\eta_t (f_{\xi_t}(x_t) - f_{\xi_t}^*)}{2} \\
&= B_\psi(x_*; x_t) - \eta_t \left(1 - \frac{1}{2}\right) \left(f_{\xi_t}(x_t) - f_{\xi_t}^*\right) + \eta_t (f_{\xi_t}(x_*) - f_{\xi_t}^*) \\
&= B_\psi(x_*; x_t) - \frac{\eta_t}{2} \underbrace{\left(f_{\xi_t}(x_t) - f_{\xi_t}^*\right)}_{\geq 0} + \eta_t (f_{\xi_t}(x_*) - f_{\xi_t}^*) \\
&\stackrel{(10)}{\leq} B_\psi(x_*; x_t) - \frac{\alpha}{2} \left(f_{\xi_t}(x_t) - f_{\xi_t}^*\right) + \eta_b (f_{\xi_t}(x_*) - f_{\xi_t}^*) \\
&= B_\psi(x_*; x_t) - \frac{\alpha}{2} \left(f_{\xi_t}(x_t) - f_{\xi_t}(x_*)\right) - \frac{\alpha}{2} \left(f_{\xi_t}(x_*) - f_{\xi_t}^*\right) + \eta_b (f_{\xi_t}(x_*) - f_{\xi_t}^*) \\
&\leq B_\psi(x_*; x_t) - \frac{\alpha}{2} \left(f_{\xi_t}(x_t) - f_{\xi_t}(x_*)\right) + \eta_b (f_{\xi_t}(x_*) - f_{\xi_t}^*)
\end{aligned}
$$

Recall from (10) that we have

$$\alpha = \min\left\{\frac{\mu_\psi}{2cL_{\max}}, \eta_b\right\} \leq \eta_t \leq \eta_b.$$

By a simple rearrangement we have

$$\frac{\alpha}{2} \left(f_{\xi_t}(x_t) - f_{\xi_t}^*\right) \leq B_\psi(x_*; x_t) - B_\psi(x_*; x_{t+1}) + \eta_b (f_{\xi_t}(x_*) - f_{\xi_t}^*).$$

Taking an expectation on both sides, dividing by $\alpha$, and applying the definition of $\sigma^2$ yields

$$\mathbb{E}\left[f(x_t) - f(x_*)\right] \leq \frac{2}{\alpha} \left(\mathbb{E}\left[B_\psi(x_*; x_t)\right] - \mathbb{E}\left[B_\psi(x_*; x_{t+1})\right]\right) + \frac{2\eta_b}{\alpha} \sigma^2.$$

Summing across time, applying convexity of $f$, and dividing by $t$ gives

$$\mathbb{E}\left[f(\bar{x}_t) - f(x_*)\right] \leq \frac{1}{t} \sum_{s=1}^{t} \mathbb{E}\left[f(x_s) - f(x_*)\right] \leq \frac{2B_\psi(x_*; x_1)}{\alpha t} + \frac{2\eta_b \sigma^2}{\alpha}.$$

$\square$

### G.2.1 Constant stepsize corollary

In this section we present the constant stepsize corollary for Theorem 7. If $\eta_b \leq \mu_\psi/2L$ then $\mathrm{mSPS_{max}}$ with $c = 1$ is a constant stepsize because of the lower bound (10), $\eta_t = \eta_b$, and we have that $\eta_b = \alpha$. Therefore plugging in these values into Theorem 7 gives the following corollary.

**Corollary 8.** *Assume $f_\xi$ is convex and $L$ smooth with respect to a norm $||\cdot||$ almost surely, assumption 1 holds, and $\psi$ is $\mu_\psi$ strongly convex over $\mathcal{X}$ with respect to the norm $||\cdot||$. Then stochastic mirror descent with $\eta \leq \mu_\psi/2L$ guarantees*

$$\mathbb{E}\left[f(\bar{x}_t) - f(x_*)\right] \leq \frac{2B_\psi(x_*; x_1)}{\eta t} + 2\sigma^2.$$

### G.3 SGD with preconditioning

In this section we extend the result of $\mathrm{mSPS_{max}}$ to the non-convex setting when $f$ is smooth and satisfies the PL condition. The result generalizes Theorem 3.6 in Loizou et al. (2021) by replacing SGD with preconditioned SGD. Note that in this case we have $\psi(x) = \frac{1}{2}\langle x, Mx \rangle$ is ($\mu_\psi = 1$)-stronlgy convex with respect to the norm $||\cdot||_M$ and $B_\psi(x; y) = \frac{1}{2}||x - y||_M^2$.

**Assumption 2** (Polyak (1964); Łojasiewicz (1963)). *Assume that $f : \mathbb{R}^n \to \mathbb{R}$ satisfies the PL condition with respect to the norm $||\cdot||_*$ if there exists $\mu > 0$ such that for all $x \in \mathbb{R}^n$*

$$||\nabla f(x)||_*^2 \geq 2\mu(f(x) - f^*). \tag{26}$$

**Theorem 9.** *Assume that $f$ and $f_\xi$ are $L$ smooth with respect to the norm $||\cdot||_M$ almost surely, where $M$ is a positive definite matrix. Furthermore, assume that $f$ satisfies the PL condition (26) with respect to the norm $||\cdot||_{M^{-1}}$, then unconstrained stochastic mirror descent with $\psi(x) = \frac{1}{2}||x||_M^2$ and stepsizes*

$$\eta_t = \min\left\{\frac{f_{\xi_t}(x_t) - f_{\xi_t}^*}{c\,||\nabla f_{\xi_t}(x_t)||_{M^{-1}}^2}, \eta_b\right\},$$

*with $c > \frac{L_{\max}}{4\mu}$ and $\eta_b < \max\left\{1/(\frac{1}{\alpha} - 2\mu + \frac{L_{\max}}{2c}), \frac{1}{2cL_{\max}}\right\}$, guarantees*

$$\mathbb{E}\left[f(x_{t+1}) - f(x_*)\right] \leq \nu^t(f(x_1) - f(x_*)) + \frac{L\sigma^2\eta_b}{2(1-\nu)c},$$

*where $\alpha = \min\{\frac{1}{2cL_{\max}}, \eta_b\}$ and $\nu = \eta_b(\frac{1}{\alpha} - 2\mu + \frac{L_{\max}}{2c}) \in (0, 1)$.*

*Proof.* We have that the algorithm performs updates of the form

$$x_{t+1} = x_t - \eta_t M^{-1} \nabla f_{\xi_t}(x_t).$$

We first apply the $L$ smoothness upper bound on $f$,

$$f(x_{t+1}) \leq f(x_t) + \langle \nabla f(x_t), x_{t+1} - x_t \rangle + \frac{L}{2}||x_{t+1} - x_t||_M^2$$

$$= f(x_t) - \eta_t \langle \nabla f(x_t), M^{-1}\nabla f_{\xi_t}(x_t) \rangle + \frac{L\eta_t^2}{2}||M^{-1}\nabla f_{\xi_t}(x_t)||_M^2$$

$$= f(x_t) - \eta_t \langle \nabla f(x_t), M^{-1}\nabla f_{\xi_t}(x_t) \rangle + \frac{L\eta_t^2}{2}\langle M^{-1}\nabla f_{\xi_t}(x_t), MM^{-1}\nabla f_{\xi_t}(x_t) \rangle$$

$$= f(x_t) - \eta_t \langle \nabla f(x_t), M^{-1}\nabla f_{\xi_t}(x_t) \rangle + \frac{L\eta_t^2}{2}||\nabla f_{\xi_t}(x_t)||_{M^{-1}}^2$$

$$\implies \frac{f(x_{t+1}) - f(x_t)}{\eta_t} \leq -\langle \nabla f(x_t), M^{-1}\nabla f_{\xi_t}(x_t) \rangle + \frac{L\eta_t}{2}||\nabla f_{\xi_t}(x_t)||_{M^{-1}}^2$$

$$\overset{(25)}{\leq} -\langle \nabla f(x_t), M^{-1}\nabla f_{\xi_t}(x_t) \rangle + \frac{L}{2c}(f_{\xi_t}(x_t) - f_i^*)$$

$$= -\langle \nabla f(x_t), M^{-1}\nabla f_{\xi_t}(x_t) \rangle + \frac{L}{2c}(f_{\xi_t}(x_t) - f_{\xi_t}(x_*)) + \frac{L}{2c}(f_{\xi_t}(x_*) - f_i^*)$$

We proceed by taking an expectation over $\xi_t$ condition on knowing $x_t$.

$$\mathbb{E}_t\left[\frac{f(x_{t+1}) - f(x_t)}{\eta_t}\right] = -\langle\nabla f(x_t), M^{-1}\nabla f(x_t)\rangle + \frac{L}{2}(f(x_t) - f(x_*)) + \frac{L}{2}\mathbb{E}_t\left[(f_{\xi_t}(x_*) - f_i^*)\right]$$

$$\leq -\|\nabla f(x)\|_{M^{-1}}^2 + \frac{L}{2c}(f(x_t) - f(x_*)) + \frac{L}{2c}\sigma^2$$

$$\overset{(26)}{\leq} -2\mu(f(x_t) - f(x_*)) + \frac{L}{2c}(f(x_t) - f(x_*)) + \frac{L}{2c}\sigma^2$$

Let $\alpha = \min\{\frac{\mu_\psi}{2cL_{\max}}, \eta_b\}$.

$$\mathbb{E}_t\left[\frac{f(x_{t+1}) - f(x_*)}{\eta_t}\right] \leq \mathbb{E}_t\left[\frac{f(x_t) - f(x_*)}{\eta_t}\right] - 2\mu(f(x_t) - f(x_*)) + \frac{L}{2c}(f(x_t) - f(x_*)) + \frac{L}{2c}\sigma^2$$

$$\leq \frac{1}{\alpha}(f(x_t) - f(x_*)) - 2\mu(f(x_t) - f(x_*)) + \frac{L}{2c}(f(x_t) - f(x_*)) + \frac{L}{2c}\sigma^2$$

$$= \left(\frac{1}{\alpha} - 2\mu + \frac{L}{2c}\right)(f(x_t) - f(x_*)) + \frac{L}{2c}\sigma^2$$

$$\leq \left(\frac{1}{\alpha} - 2\mu + \frac{L_{\max}}{2c}\right)(f(x_t) - f(x_*)) + \frac{L}{2c}\sigma^2$$

Therefore we have the following sequence of inequalities,

$$\mathbb{E}_t\left[\frac{f(x_{t+1}) - f(x_*)}{\eta_b}\right] \leq \mathbb{E}_t\left[\frac{f(x_{t+1}) - f(x_*)}{\eta_t}\right] \leq \left(\frac{1}{\alpha} - 2\mu + \frac{L_{\max}}{2c}\right)(f(x_t) - f(x_*)) + \frac{L}{2c}\sigma^2$$

By the tower property of expectations and multiplying both sides by $\eta_b$ we have

$$\mathbb{E}\left[f(x_{t+1}) - f(x_*)\right] \leq \underbrace{\eta_b\left(\frac{1}{\alpha} - 2\mu + \frac{L_{\max}}{2c}\right)}_{\nu}\mathbb{E}\left[(f(x_t) - f(x_*))\right] + \frac{\eta_b L}{2c}\sigma^2.$$

If $\nu \in (0, 1)$ then iterating the inequality and summing the geometric series gives the result,

$$\mathbb{E}\left[f(x_{t+1}) - f(x_*)\right] \leq \nu^t(f(x_1) - f(x_*)) + \sum_{s=0}^{t-1}\nu^s\frac{\eta_b L}{2c}\sigma^2$$

$$\leq \nu^t(f(x_1) - f(x_*)) + \frac{\eta_b L\sigma^2}{2(1-\nu)c}.$$

Therefore, it remains to show that $0 < \nu < 1$. For the lower bound notice that $\alpha \leq \frac{1}{2cL_{\max}}$,

$$\nu = \eta_b\left(\frac{1}{\alpha} - 2\mu + \frac{L_{\max}}{2c}\right)$$

$$\geq \eta_b\left(2cL_{\max} - 2\mu + \frac{L_{\max}}{2c}\right)$$

$$= \eta_b\left(\left(2c + \frac{1}{2c}\right)L_{\max} - 2\mu\right) > 0.$$

Following similar arguments made in Loizou et al. (2021)[Theorem 3.6], we can show $\nu < 1$ by considering two cases. Recall from our assumptions we have $c > \frac{L_{\max}}{4\mu}$ and $\eta_b < \max\left\{1/(\frac{1}{\alpha} - 2\mu + \frac{L_{\max}}{2c}), \frac{1}{2cL_{\max}}\right\}$, therefore

we consider the two following cases:

$$\eta_b < \frac{1}{2cL_{\max}} \tag{27}$$

$$\eta_b < \frac{1}{\left(\frac{1}{\alpha} - 2\mu + \frac{L_{\max}}{2c}\right)}. \tag{28}$$

For the first case (27) we have $\alpha = \eta_b$ and

$$
\begin{aligned}
\nu &= \eta_b \left( \frac{1}{\eta_b} - 2\mu + \frac{L_{\max}}{2c} \right) \\
&= 1 - 2\eta_b\mu + \frac{L_{\max}}{2c}\eta_b \\
&\overset{(c > \frac{L_{\max}}{4\mu})}{<} 1 - 2\mu\eta_b + 2\mu\eta_b = 1.
\end{aligned}
$$

For the second case (28) we have $\alpha = \frac{1}{2cL_{\max}}$ and by the upper bound we have

$$\nu = \eta_b \left( \frac{1}{\alpha} - 2\mu + \frac{L_{\max}}{2c} \right) < 1.$$

However, we also have $\alpha = \frac{1}{2cL_{\max}} \le \eta_b$, to avoid a contradiction we need

$$\frac{1}{2cL_{\max}} < \frac{1}{\frac{1}{\alpha} - 2\mu + \frac{L_{\max}}{2c}} = \frac{1}{2cL_{\max} - 2\mu + \frac{L_{\max}}{2c}}.$$

Which holds by assumption since $c > \frac{L_{\max}}{4\mu}$. $\qquad\square$

## H Experiment details

In this section we provide details for our experiments including the updates for different mirror descent algorithms. Note that in all our experiments we have $f_i^* = 0$.

### H.1 Compute resources

We ran around a thousand experiments using an internal cluster, where each experiment uses a single NVIDIA Tesla P100 GPU, 40GB of RAM, and 4 CPUs. Some experiments like the synthetic ones took only few minutes to complete, while the deep learning experiments like CIFAR10 took about 12 hours.

### H.2 Mirror descent across p-norms

We select $\psi(x) = \frac{1}{2}\|x\|_p^2$ and $\mathcal{X} = \mathbb{R}^d$ for $1 < p \le 2$. We have in this case that $\psi$ is $\mu_\psi = (p-1)$ strongly convex with respect to the norm $\|\cdot\|_p$ with dual norm $\|\cdot\|_q$ where $q$ is such that $1/p + 1/q = 1$ (Orabona, 2019). Therefore, as defined in Corollary 8, $\text{mSPS}_{\max}$ with $c = 1$ is

$$\eta_t = \min \left\{ \frac{(p-1)(f_i(x_t) - f_i^*)}{\|\nabla f_i(x_t)\|_q^2}, \eta_b \right\},$$

and similarly for mSPS.

The closed form update for mirror descent in this case is given by the following coordinate wise updates (Duchi, 2018): let $\phi^p : \mathbb{R}^d \to \mathbb{R}^d$ with component functions $\phi_i^p(x) = (\|x\|_p)^{2-p} \operatorname{sign}(x_i)|x_i|^{p-1}$, then the mirror descent update with stepsize $\eta_t$ is

$$x_{t+1} = \phi^q(\phi^p(x_t) - \eta_t \nabla f_i(x_t)).$$

### H.3 Projected gradient descent with positive constrains

We select $\psi(x) = \frac{1}{2}\|x\|_2^2$ to recover projected gradient descent, in this case $\psi$ is $\mu_\psi = 1$ strongly convex with respect to the Euclidean norm and $\|\cdot\|_* = \|\cdot\|_2$. Since $\mathcal{X} = \mathbb{R}_+^d$, the non-negative orthant, the projection step amounts to clipping negative values (setting them to zero).

### H.4 Exponentiated gradient with $\ell_1$ constraint

We consider the case of supervised learning with constraint set $\mathcal{X} = \{x : \|x\|_1 \leq \lambda\}$. In our experiments we set $\lambda = 10,000 \cdot d$. To consider the exponentiated gradient algorithm we equivalently write the set $\mathcal{X}$ as a convex hull of its corners, $\mathcal{X} = \{\Lambda x : x \in \Delta_{2d}\}$ where $\Delta_{2d}$ is the $(2d-1)$-dimensional probability simplex and $\Lambda$ is a matrix with $2d$ columns and $d$ rows,

$$\Lambda = \begin{bmatrix} \lambda & -\lambda & 0 & 0 & \cdots & 0 & 0 \\ 0 & 0 & \lambda & -\lambda & \cdots & 0 & 0 \\ 0 & 0 & 0 & 0 & \cdots & 0 & 0 \\ \vdots & \vdots & \vdots & \vdots & \cdots & \vdots & \vdots \\ 0 & 0 & 0 & 0 & \cdots & \lambda & -\lambda \end{bmatrix}.$$

Therefore we can use the exponentiated algorithm with constraint set $\Delta_{2d}$ by selecting $\psi(x) = \sum_{i=1}^{2d} x_i \log(x_i)$. In this case $\psi$ is $\mu_\psi = 1$ strongly convex on $\Delta_{2d}$ with respect to the norm $\|\cdot\|_1$. Since the dual norm $\|\cdot\|_* = \|\cdot\|_\infty$ we have that $\mathrm{mSPS}_{\max}$ with $c = 1$ is

$$\eta_t = \min\left\{\frac{(f_i(x_t) - f_i^*)}{\|\nabla f_i(x_t)\|_\infty^2}, \eta_b\right\},$$

and similarly for mSPS.

The mirror descent update then can be written in two steps (Bubeck, 2015),

$$y_{t+1} = x_t \odot \exp(-\eta_t \nabla f_i(x_t))$$
$$x_{t+1} = \frac{y_{t+1}}{\|y_{t+1}\|_1}.$$

Where $\odot$ and exp are component wise multiplication and component wise exponentiation respectively.

### H.5 Additional Results across p-norms

We observe in Figure 4 that mSPS outperforms a large grid of step-sizes for most values of $p$. Note that we used the mushrooms dataset with the kernel bandwidth selected in Vaswani et al. (2019b) which satisfies interpolation.

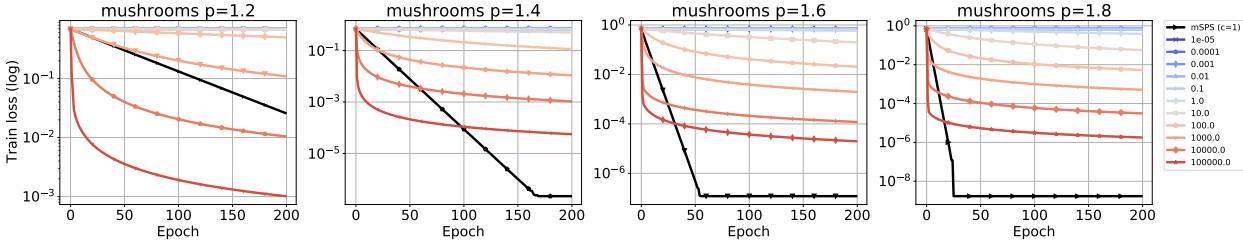

Figure 4: Comparison between mSPS with $c = 1$ and constant step-sizes on convex binary-classification problem on the mushroom dataset.

For the non-convex multi-class classification problem in Figure 5 we use MNIST. MNIST has a training set consisting of 60k examples and a test set of 10k examples. We use a 1 hidden-layer multi-layer perceptron (MLP) of width 1000. We also observe that mSPS is either competitive or better than most constant stepsizes across various values of $p$.

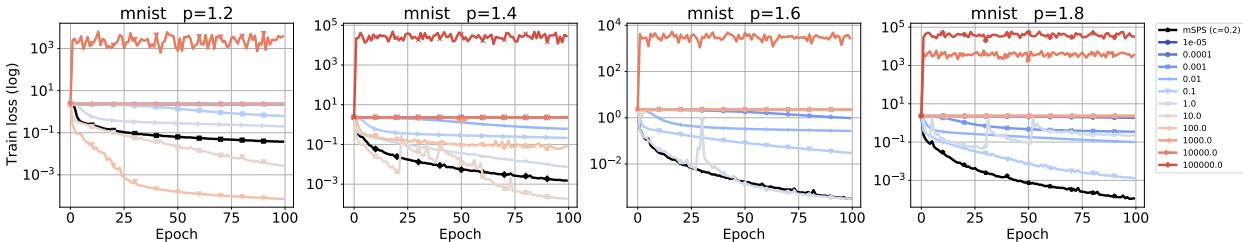

Figure 5: Comparison between mSPS with $c = .2$ and constant step-sizes on MNIST across different values of $p$.

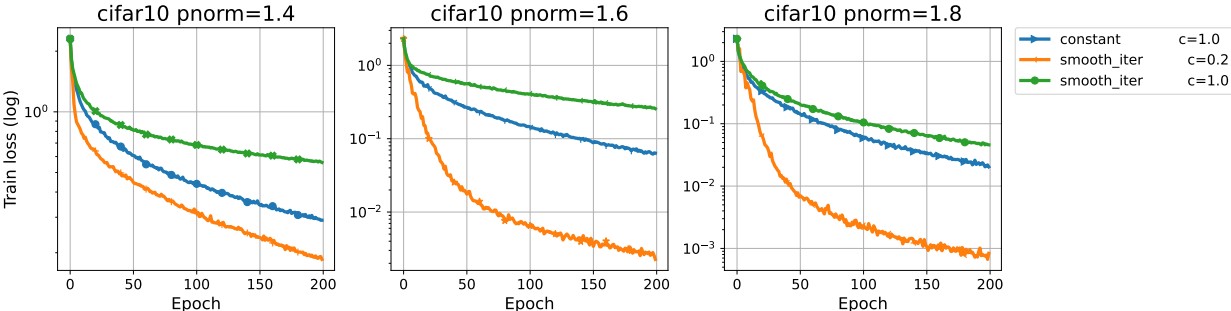

Figure 6: Comparison between mSPS with and without smoothing on non-convex multiclass classification with deep networks across different values of $p$.

