# OpenReview forum: "Stochastic Mirror Descent: Convergence Analysis and Adaptive Variants via the Mirror Stochastic Polyak Stepsize"
_TMLR — Accepted by TMLR_

### Review · Reviewer_qw5m · 2023-06-19

**Summary Of Contributions:**

This paper studies the mirror descent method for stochastic optimization. The main contributions are two-fold: on the one hand it proves new theory for constant step-size mirror descent based on an interpolation constant. On the other hand, it proposes a new method which is reminiscent of Polyak's step size adapted to the setting of mirror descent (mSPS). For this new method, theoretical results (mainly for the convex case) are established; experiments show that mSPS achieves good convergence with less tuning due to an adaptive step size.


**Audience:**

Yes

**Broader Impact Concerns:**

None.

**Claims And Evidence:**

Yes

**Requested Changes:**

Some of the below are questions which I had while reading - it might be sufficient to clarify them during the discussion period without changing the paper.

On the theoretical part:

* Do you assume that $\mathcal{X} \subset \mathcal{D}$? This seems to be somewhat implictly assumed in Def. 2 as $B_{\psi}$ is defined via $\mathcal{D}$, but I couldn't find it in the paper.

* I think you need to assume that $B_{\psi}$ is non-negative (e.g. for Corollary 2) i.e. $\psi$ is convex. I couldn't find this assumption in the paper, but maybe I just skipped it. Can you clarify?

* Page 3: what do you mean with "bounded constraint"?

* For the theory of section 6, you use the interpolation constant $\sigma^2$ and not $\sigma^2_{\mathcal{X}}$. I think this comes from the lower bound of $\eta_t$, but it seemed a bit hidden to me why this is the case - I guess that one would prefer to have $\sigma^2_{\mathcal{X}}$ if it was possible. Maybe you could explain this phenomenon in a short sentence/paragraph.


On the experiments:

* For some experiments, it looks like the biggest step size is still the best one for mirror descent e.g. first row or cifar10, p=1.4. I would have expected that the largest one is always too large so that we can be sure that the best step size is among the ones that were run. But maybe its just misleading because there are lines missing from diverged runs? If not, I think you should extend the step size range in order to make sure that the (approximately) best step size is among the ones that were run.

* The runs for ijcnn and rcv1 for the l1-constrained problem look a bit strange: it looks to me like mirror descent would need a smaller step size as the ones that are displayed are already quite unstable. Also the scale of the loss there (in the orders of 10^6) looks uncommonly high to me. Related to this I did not find the value of $\lambda$ you chose. Further, in section G.4 in the mirror descent update I can't see any influence of $\lambda$ - but it should have. I think the result from Bubeck, 2015 is only valid if the domain is the simplex, but in your case we have an additional $\Lambda$?

* I understand that you use the smoothing of $\eta_b$ and the values of $c$ for mSPS based on the Loizou et al 2021 paper. However, we should be careful as this introduces additional tuning for mSPS which has to be accounted for - in the sense that some hyperparameters are varied per experiment which is not done for Mirror Descent. It is unclear to me when $c=1$ or when $c=0.2$ should be used - is it only depending on convexity and what's the intuition for this? For completeness, it would be nice to see - for example - also the results for the choice $c=1$ in the experiments where $c=0.2$ is used.

Minor remarks:

* In several spots, you write $f_i$ instead of $f_\xi$ (e.g. page 2, end of first bullet or several times in Appendix F).

* You cite an Assumption right below Assum. 1, from the book of Orabona. As this book has lots of content, it would be nice to be more specific where that assumption can be found in the book. The same applies to the result of Hanzely & Richtarik 2021 and Dragomir et al 2021 in section 5.3. Please state explicitly which theorem you compare to as it makes it easier for the reader.

**Strengths And Weaknesses:**

Strengths:

Adaptive learning rates such as the Polyak step size have drawn interest in the recent years because of their favourable performance for problems with (almost) interpolation and their improved sensitivity to learning rate tuning.
This paper studies the important question how we can make use of these techniques for mirror descent methods where specific variable constraints or domains motivate the use of a different proximal term.

Even though the results of section 5 and 6 seem somewhat unrelated at first sight, the paper has a clear structure and the contributions of these two sections are linked through their relation to the interpolation constant. The paper gives a thorough literature review and relates their results nicely within the context of previous work which makes the paper an enjoyable read.


Weaknesses:

* I think that the result of section 5 (e.g. Theorem 1) is useful mainly in the cases where the interpolation constant is reasonably small. A major difference to the works compared in the paper is that one can not force the constant term small by making the step size small (even though the other terms are large). For the result of e.g. Dragomir et al 2021 this is different - the constant term in their work multiplies with the step size. Hence, if the interpolation constant is not very small, the constant term in Thm. 1 is not in control of the user. While this is in general fine for me, I think that it needs to be pointed out and explained clearer as it limits the applicability of the convergence result to problems with zero or reasonably small interpolation constant.

* Regarding the point above, I think that the example in 5.3.1 is insightful to showcase the advantage of your results but also slightly one-sided. You could give a second example where the interpolation constant is large to show what happens in such a case (I would not see it as a weakness if your result is less useful for such settings).

* The same remark applies to Theorem 5 and subsequent results: if $\eta_b$ is set small then at some point $\alpha=\eta_b$ and it will cancel out in the constant term. Hence, the constant term can not be made small by reducing the parameter $\eta_b$.

* The theoretical results (except for Thm 9 using PL condition) are mainly in the convex setting. From the current paper it is unclear to me if the presented results could be extended to more general nonconvex settings.

---

> ### Author Response · Authors · 2023-08-02
> **Author Response**
>
> Thank you for your review! We appreciate your deep understanding of our work and helpful comments.
>
> > I think that the result of section 5 (e.g. Theorem 1) is useful mainly in the cases where the interpolation constant is reasonably small. [...]  I think that it needs to be pointed out and explained clearer as it limits the applicability of the convergence result to problems with zero or reasonably small interpolation constant.
>
> Although we provide convergence for a larger class of problems and methods than the related work, we agree that our analysis does not allow one to control the neighborhood with a smaller stepsize as in Dragmoir et. al.. We have updated Section 5.3 accordingly to reflect this limitation.
>
> > Regarding the point above, I think that the example in 5.3.1 is insightful to showcase the advantage of your results but also slightly one-sided. You could give a second example where the interpolation constant is large to show what happens in such a case (I would not see it as a weakness if your result is less useful for such settings).
>
> Example 5.3.1 was created not only to showcase our results but also interpolation in the constrained optimization setting where the optimum may not have zero gradient or zero variance. We agree it is important to address the limitations of our results at the other extreme (far from interpolation) and hope that our modification to section 5.3 addresses this issue.
>
> > The theoretical results (except for Thm 9 using PL condition) are mainly in the convex setting. From the current paper it is unclear to me if the presented results could be extended to more general nonconvex settings.
>
> In general, we agree that studying stochastic mirror descent in the non-convex setting with interpolation is of interest and would be important future work. Our Theorem 3 does not assume $f$ to be convex and could be used in some non-convex settings. However, it is out of scope for the present submission.
>
> > Do you assume that $\mathcal{X} \subseteq\mathcal{D}$?
>
> Yes, we assume that the domain $\mathcal{D}$ of the mirror map $\psi$ includes the constraint $\mathcal{X}$. We state this in Assumption 1, however, we have updated the assumption to clearly state that the symbol $\mathcal{D}$ is the domain of $\psi$.
>
> > I think you need to assume that is non-negative (e.g. for Corollary 2) i.e. $\psi$ is convex.
>
> Thank you for spotting this. We assume $\psi$ is convex throughout the paper via Lemma 1. We agree this assumption is not clearly stated for Section 5. We will add it to assumption 1.
>
> > Page 3: what do you mean with "bounded constraint"?
>
> We mean that the constraint set $\mathcal{X}$ is bounded.
>
> > For the theory of section 6, you use the interpolation constant $\sigma^2$ and not $\sigma^2_{\mathcal{X}}$. I think this comes from the lower bound of $n_t$ [...] Maybe you could explain this phenomenon in a short sentence/paragraph.
>
> Indeed this is one of the challenges. The lower bound actually does not need to hold in the constraint case as the gradient at the optimum need not be zero and hence would violate the lower bound. Generalizing this stepsize and lower bound with a different quantity than the norm of the gradient would be interesting future work. We will add a statement in the paper.

---

> > ### Author Response · Authors · 2023-08-02
> > **Author Response Continued**
> >
> > > For some experiments, it looks like the biggest step size is still the best one for mirror descent e.g. first row or cifar10, p=1.4. I would have expected that the largest one is always too large so that we can be sure that the best step size is among the ones that were run. But maybe its just misleading because there are lines missing from diverged runs?
> >
> > For all experiments we swept across  $\{10^{-5}, 10^{-4}, 10^{-3}, 10^{-2}, 10^{-1}, 1, 10^{1},10^{2},10^{3},10^{4},10^{5}\}$. Indeed, the larger stepsizes for $p=1.4$ in cifar10 had diverged. For the first row, we will include new results up to $10^8$.
> >
> > > The runs for ijcnn and rcv1 for the l1-constrained problem look a bit strange: it looks to me like mirror descent would need a smaller step size as the ones that are displayed are already quite unstable. Also the scale of the loss there (in the orders of 10^6) looks uncommonly high to me. Related to this I did not find the value of you chose. Further, in section G.4 in the mirror descent update I can't see any influence of $\lambda$ but it should have. I think the result from Bubeck, 2015 is only valid if the domain is the simplex, but in your case we have an additional $\Lambda$?
> >
> > Similar to our cifar experiments we will include additional results with stepsizes as small as $10^{-8}$.
> >
> > We can use the exponentiated gradient algorithm since we reduce the problem from a $\ell_1$ constraint  to a simplex constraint by rewriting  the $\ell_1$ constraint as $\mathcal{X} = \Lambda x$ where $x$ is constrained to the simplex. Therefore, a loss $f$ defined over $\mathcal{X}$ becomes a new loss $g(x) = f(\Lambda x)$ with a simplex constraint on $x$. The influence of $\lambda$ is given by the matrix $\Lambda$ as defined in section G.4. This reduction is standard as described in [1]. For our experiments we used a value of $10,000 * d$ for $\lambda$ where $d$ is the number of features. We've added the detail to section G.4.
> >
> > >I understand that you use the smoothing of $\eta_b$ and the values of $c$ for mSPS based on the Loizou et al 2021 paper.
> >
> >  We only used smoothing for the non-convex deep learning experiments (Figure 3). In the other experiments we simply used $mSPS_{\max}$ with $c =1$ as prescribed by the smooth convex theory
> >
> >
> > > It is unclear to me when $c=1$ or $c=0.2$ should be used  - is it only depending on convexity and what's the intuition for this? For completeness, it would be nice to see - for example - also the results for the choice in the experiments where is used.
> >
> > In general, the parameter $c$ depends on whether the loss is strongly convex or smooth and convex. Outside of the convex case, Loizou et. al 2021 suggested $c=0.2$ for deep learning with smoothed $SPS_{\max}$ and in general seems to perform better than $c=1$.
> >
> > > In several spots, you write $f_i$ instead of $f_\xi$ (e.g. page 2, end of first bullet or several times in Appendix F).
> >
> > We use $f_i$ to explicitly refer to the finite-sum case and introduce this notation at the beginning of paragraph 2. The reference in page 2 is for finite-sum but Appendix F should be $f_\xi$ more generally. We have updated the appendix accordingly.
> >
> > > it would be nice to be more specific where that assumption can be found in the book. The same applies to the result of Hanzely & Richtarik 2021 and Dragomir et al 2021 in section 5.3.
> >
> > Thank you for the suggestion, we will add the numerical references for better readability.
> >
> >
> > [1] Schuurmans, Dale, and Martin A. Zinkevich. "Deep learning games." Advances in neural information processing systems 29 (2016).

---

> > > ### Comment · Reviewer_qw5m · 2023-08-09
> > > **Response**
> > >
> > > Dear authors,
> > >
> > > thank you for your detailled response and sorry for my rather late answer. In general, I think that my concerns have been adressed, except for the following: On the point of how to choose $c$, as you say that $c=0.2$ works better than $c=1$ in the nonconvex case, then this is exactly what I was concerned about. If improved performance or robustness can only be achieved by adding an additional parameter, whose value has to be set correctly to get good performance, this partially defeats the original purpose. So I would still be interested in seeing how much worse $c=1$ without smoothing performs in the nonconvex experiments. Can you add these to the paper?
> > >
> > > Further, even though the Loizou et al. paper suggested these choices, it remains unclear to me why it would depend only on the class of the problem (convex or smooth/nonconvex): the theory does not really indicate this (as $c \geq 1/2$ is required in the convex case, but their Thm. 3.8 also requires a lower bound on $c$ in the nonconvex case, whose value is hard to determine). Hence, I am not fully convinced that the choice of $c$ can only be made based on whether it is a convex problem or not.
> > >
> > > **A general remark:** In your updated PDF, as you didn't mark the changes in color, it was hard to spot the changes you made.

---

> > > > ### Author Response · Authors · 2023-08-09
> > > > **Response on the value of $c$**
> > > >
> > > > > I would still be interested in seeing how much worse $c=1$ without  without smoothing performs in the nonconvex experiments. Can you add these to the paper?
> > > >
> > > > We are happy to add these experiments to the paper. Indeed, the original paper does not have strong theoretical justifications for why $c=0.2$ should work, empirically they have noticed that "any value of c ≥ 0.2 results in convergence. In this case, [they] observed that across models and datasets, the fastest convergence is obtained with c = 0.2." Despite non-convex theoretical results being out of scope we are happy to provide more experiments and evidence in the SMD case for this choice of $c$ and pose it as an open question.

---

### Review · Reviewer_Uiqa · 2023-06-27

**Summary Of Contributions:**

This paper considers stochastic mirror descent with both constant step sizes and Polyak stepsizes. For the constant step sizes, the paper presents convergence in a neighborhood for strongly convex problems, and convergence in terms of Bregman distance of $f$ for smooth problems. For the Polyak step sizes, the paper presents convergence in a neighborhood for strongly convex problems, and convergence in function values for smooth and convex problems. Experimental results are presented to show the effectiveness of the proposed algorithm.

**Audience:**

Yes

**Broader Impact Concerns:**

I have no impact concerns.

**Claims And Evidence:**

Yes

**Requested Changes:**

The paper is very well written and the results are solid. My only concern is the novelty of the analysis. Both the analysis and the results are a bit standard. I would like to see more discussions on the novelty of the paper in the revision, e.g., what are the technical challenges of the extension and technical contributions of the analysis?

Minor comments:
Section 3.1: "it it" should be "it"
Section 5.3.1: "is a quadratic" should be "is quadratic"
Section C.1: "stronly" should be "strongly"

**Strengths And Weaknesses:**

**Strength**

- Mirror descent is a powerful extension of gradient descent which covers many instantiations of existing algorithms, e.g., exponentiated gradient, gradient descent. The paper studies convergence rate of stochastic mirror descent. The proposed results are general.

- The existing results often require a bounded sub-gradient assumption or a bounded variance assumption. The paper gives convergence rates without these assumptions.

- The existing analysis on Polyak step size is conducted for SGD. The paper gives the first analysis on Polyak step size for stochastic mirror descent. The analysis seems to be rigorous.

- There are various experimental results.

**Weakness**

- While mirror descent is more general, the framework of studying the convergence of mirror descent has already been established in the literature. The analysis presented in the paper seems to be standard in the mirror descent literature. For example, all the analysis is based on Lemma 1, which has already been developed in the literature.

- Within the recent progress in optimization, people have already known how to handle the unbounded gradient assumption. For example, it is now becoming standard to remove the bounded gradient assumption by using the smoothness assumption of the loss function. Therefore, it is not surprising to remove the bounded gradient assumption in the paper.

- The analysis of Polyak step sizes has already been done for SGD. The paper extends it to SMD. While this extension is definitely useful, it seems that the extension follows standard techniques.

---

> ### Author Response · Authors · 2023-08-02
> **Author Response**
>
> Thank you for your review!
>
> We appreciate you finding our results useful and our paper well-written.
>
> > The analysis presented in the paper seems to be standard in the mirror descent literature. For example, all the analysis is based on Lemma 1, which has already been developed in the literature.
>
> We agree Lemma 1 is a standard starting point for analyzing mirror descent, and you will see it used in countless papers, however we do not see this as a weakness.  Our contribution to the community and novelty stems from our results themselves. We establish **novel** convergence guarantees for stochastic mirror descent in various settings. For example, as we emphasized in the paper our results are the only results that guarantee the convergence of the exponentiated algorithm under interpolation, and we have done so using (1) a constant stepsize and relative smoothness, and (2) an adaptive stepsize.
>
> Furthermore, from an analysis perspective, Lemma 1 is a **starting** point for which we then incorporate other ideas and assumptions such as relative smoothness or interpolation, all of which contribute to a novel analysis as a whole.
>
>
> > Within the recent progress in optimization, people have already known how to handle the unbounded gradient assumption. For example, it is now becoming standard to remove the bounded gradient assumption by using the smoothness assumption of the loss function. Therefore, it is not surprising to remove the bounded gradient assumption in the paper.
>
> We agree that smoothness has become useful in removing bounded variance or gradient assumptions in stochastic unconstrained optimization. However, it is not a standard tool yet  in constrained optimization or with relative smoothness. With just two existing works, [1] with Euclidean assumptions and the Frank-Wolfe algorithm and another [2] using mirror descent and relative smoothness. Our work has actually provided new results on both fronts.  In the constrained Euclidean case and compared to [1] we allow for convergence without assuming exact interpolation and we can achieve exact convergence without requiring zero variance at the optimium (see Section 2 paragraph "Interpolation in constrained optimization" for discussion).
> In Section 5.3 we provide an in-depth comparison to [2], where we explain our novel convergence guarantees under relative smoothness.
>
>
> [1] Xiao, Tesi, Krishnakumar Balasubramanian, and Saeed Ghadimi. "Improved complexities for stochastic conditional gradient methods under interpolation-like conditions." Operations Research Letters 50.2 (2022): 184-189.
>
> [2] Dragomir, Radu Alexandru, Mathieu Even, and Hadrien Hendrikx. "Fast stochastic bregman gradient methods: Sharp analysis and variance reduction." International Conference on Machine Learning. PMLR, 2021.

---

> > ### Author Response · Authors · 2023-08-02
> > **Author Response Continued**
> >
> > > The analysis of Polyak step sizes has already been done for SGD. The paper extends it to SMD. While this extension is definitely useful, it seems that the extension follows standard techniques.
> >
> > Thank you for finding our extension useful, and hope others will as well. Regarding the standard techniques, we have put much effort in making our analysis readable and easy to follow. In this case it is true that our adaptive results follow much of the original paper, however, we had to first figure out (1) what is the appropriate generalization of the stepsize for the analysis to work with mirror descent and (2) what other different results  and assumptions are required. For everything to fall into place nicely we needed to generalize the stepsize by replacing the Euclidean norm with a general dual norm $|| \cdot||_\ast$ that is dual to the norm for which $\psi$ is strongly convex. In hindsight this may be obvious especially given our presentation and motivation of the stepsize via the well-known Lemma 1, but this should not diminish the novelty of our results and generalization. Furthermore, we also reinforce the importance of these results, despite the similarity of the previous work, by demonstrating improvements if the mirror map is chosen appropriately (see Section 6.2 "Comparison with SPS" paragraph).
> >
> > >  I would like to see more discussions on the novelty of the paper in the revision, e.g., what are the technical challenges of the extension and technical contributions of the analysis?
> >
> > As we have mentioned above, our novelty mainly lies in our results which we believe to be of interest to the community. To summarize we have novel guarantees for the convergence of stochastic mirror descent under interpolation with conditions more general than existing works and including cases not previously studied such as:
> >  - the exponentiated gradient algorithm
> >  - a new adaptive stepsize which measures gradients via a dual norm $|| \cdot||_\ast$.
> >
> > We kindly point out that the [TMLR guidelines for acceptance](https://jmlr.org/tmlr/acceptance-criteria.html) encourages the publication of interesting results without emphasis on novelty of methodology. More importantly, our analysis does incorporate  new ideas that differ from standard analysis and hence provides a novelty in and of itself. We are happy to improve the clarity of contributions in our next revision with the details from this rebuttal.

---

### Review · Reviewer_eT7G · 2023-07-25

**Summary Of Contributions:**

The paper investigates the convergence of stochastic mirror descent (SMD) under interpolation in relatively smooth and smooth convex optimization. The authors provide new convergence guarantees for SMD with a constant stepsize in relatively smooth convex optimization. They also propose a new adaptive stepsize scheme, the mirror stochastic Polyak stepsize (mSPS), for smooth convex optimization. Notably, their convergence results do not make bounded gradient assumptions or bounded variance assumptions, and they show convergence to a neighborhood that vanishes under interpolation.

**Audience:**

No

**Broader Impact Concerns:**

In my opinion, there don't appear to be any specific concerns regarding Broader Impact.

**Claims And Evidence:**

Yes

**Requested Changes:**

The use of Polyak'stepsize seems to be of limited use. Maybe I missed something, is it possible to use the adaptive stepsize when the optimal value is unknown? can you provide a discussion?

**Strengths And Weaknesses:**

The paper is well-structured and provides a comprehensive analysis of the SMD under different conditions. The introduction of Polyak as an adaptive stepsize for SMD seems to be an interesting contribution, and the authors' approach to not using an online to batch reduction is noteworthy. The paper also stands out for its extensive numerical experiments for adaptive SMD across a wide variety of domains and mirror descent algorithms for both constrained and unconstrained problems.


In terms of technical aspects, the analysis of relative smoothness and Polyak step size appears to be fairly standard, and the convergence results are as expected. Additionally, it's unclear to me what the standard for TMLR is, but in my view, this paper may not meet the high standards set by conferences such as ICML or NeurIPS.

---

> ### Author Response · Authors · 2023-08-02
> **Author Response**
>
> Thank you for your review and for finding our paper both comprehensive and interesting.
>
> > In terms of technical aspects, the analysis of relative smoothness and Polyak step size appears to be fairly standard, and the convergence results are as expected.
>
> Although we use standard tools from the mirror descent literature, we have combined these tools with new assumptions to yield novel results. For example, our work establishes the first exact convergence results for the exponentiated gradient algorithm under interpolation with both a constant and adaptive stepsize.
>
> We believe our paper to be of great importance to the mirror descent and constrained optimization community as we have provided:
>
> - a new definition of interpolation appropriate for constrained optimization (see paragraph "Interpolation in constrained optimization" in Section 2 for more details)
> - novel convergence guarantees under our new constrained interpolation condition for SMD
> - introducing the classic polyak stepsize to mirror descent with comprehensive analysis and experiments to showcase its usefulness in machine learning applications
>
> > The use of Polyak'stepsize seems to be of limited use. Maybe I missed something,  is it possible to use the adaptive stepsize when the optimal value is unknown?
>
> Indeed the Polyak stepsize does require knowledge of the optimal value. However, the stochastic version (SPS) and our mirror descent version (mSPS) only requires knowledge of the optimal value for the sampled function $f_{\xi}$. In many machine learning applications like supervised learning such a value is usually known and is typically zero. In supervised learning, $f_\xi^\ast$ is the best possible loss a model can have on one example. In Section 5.2.1 we also discuss the important problem of finding stationary distributions where $f_\xi^\ast = 0$; our analysis establishes exact convergence to the stationary distribution for the stochastic exponentiated algorithm. In general it would be interesting to extend our work to the case where $f_\xi^\ast$ is not known and must be approximated such as in [1].
>
> [1] Orvieto, Antonio, Simon Lacoste-Julien, and Nicolas Loizou. "Dynamics of sgd with stochastic polyak stepsizes: Truly adaptive variants and convergence to exact solution." Advances in Neural Information Processing Systems 35 (2022): 26943-26954.

---

### Decision · Action_Editors · 2023-09-07

**Recommendation:** Accept with minor revision

**Comment:**

In consensus with the reviewers, I find this paper to have enough novelty and significance to warrant publication, so I am happy to recommend publication.  It's well-written and pushes existing results forward in a few new directions.


Please make a few small changes, e.g., some of the authors in the bibliography have abbreviated first names (A. Nemirovski...) while most others don't; please update references to journal versions when possible; and I'd change the "Anatoli Iouditski" and Nesterov paper to the "Juditsky" transliteration (as he himself uses on his PDF copy, https://arxiv.org/pdf/1401.1792.pdf ).

Lemma 3, the three point property, was proven by Chen and Teboulle, "Convergence analysis of a proximal-like minimization algorithm using Bregman functions", 1993, so I would cite that instead of Bubeck '15/Orabona '19.

One reviewer requested a link to an open source implementation of the experiments, if you have that.

Reviewer qw5m had requested that the case c=1 to be run for the non-convex experiments (in Fig. 3). If this doesn't fit in well with the main paper, it would be OK to put it in the appendix.

Finally, to make the paper more accessible to a general audience, I would add a small section on relative smoothness to the appendix, since it's a new concept (from Lu, Freund & Nesterov '16)

**Audience:**

Mirror descent is a specialized algorithm, though increasingly well-known. As it is quite general, it is interesting theoretically, and it has some cases where it is "better" than standard gradient descent (e.g., reducing the dependence on dimension), although it is arguably still not a workhorse algorithm in the toolkit of practitioners.  So I think work on mirror descent has a sizable audience, and work such as this one will help to increase its applicability and hence audience.

Furthermore, the stochastic setting is also of very general interest since it covers training problems for machine learning.  Work such as this one that propose adaptive stepsizes are of great interest, since finding the right "learning rate" is a major issue for machine learning.

Overall, this paper will have some readers interested in the theory, and some interested in using the algorithms. I find it to have at least a typical audience size for a TMLR paper.

**Claims And Evidence:**

The authors provide a new analysis of stochastic mirror descent with an adaptive stepsize (borrowed from recent SGD literature) and under new technical assumptions, such as weaker assumptions on the noise. The analysis is proven rigorously.  They also demonstrate some of the results with numerical experiments.  The reviewers did not find any reasons to doubt the claims.